# Learning Pseudorandom Numbers with Transformers: Permuted Congruential Generators, Curricula, and Interpretability

**Tao Tao**
Department of Physics, University of Maryland, College Park, USA
`tao2021@umd.edu`

**Maissam Barkeshli**
Department of Physics, University of Maryland, College Park, USA
Meta FAIR
Joint Quantum Institute, University of Maryland
`maissam@umd.edu`

## Abstract

We study the ability of Transformer models to learn sequences generated by Permuted Congruential Generators (PCGs), a widely used family of pseudo-random number generators (PRNGs). PCGs introduce substantial additional difficulty over linear congruential generators (LCGs) by applying a series of bit-wise shifts, XORs, rotations and truncations to the hidden state. We show that Transformers can nevertheless successfully perform in-context prediction on unseen sequences from diverse PCG variants, in tasks that are beyond published classical attacks. In our experiments we scale moduli up to $2^{22}$ using up to 50 million model parameters and datasets with up to 5 billion tokens. Surprisingly, we find even when the output is truncated to a single bit, it can be reliably predicted by the model. When multiple distinct PRNGs are presented together during training, the model can jointly learn them, identifying structures from different permutations. We demonstrate a scaling law with modulus $m$: the number of in-context sequence elements required for near-perfect prediction grows as $\sqrt{m}$. For larger moduli, optimization enters extended stagnation phases; in our experiments, learning moduli $m \geq 2^{20}$ requires incorporating training data from smaller moduli, demonstrating a critical necessity for curriculum learning. Finally, we analyze embedding layers and uncover a novel clustering phenomenon: the top principal components spontaneously group the integer inputs into bitwise rotationally-invariant clusters, revealing how representations can transfer from smaller to larger moduli.

## 1 Introduction

Transformer-based models have achieved remarkable success across language, vision, and algorithmic tasks, demonstrating an ability to capture complex patterns from large-scale data (Vaswani et al., 2023; Dosovitskiy et al., 2021). Beyond supervised training, they can acquire new behaviors directly from examples provided in the input, a phenomenon known as in-context learning (Brown et al., 2020; Olsson et al., 2022). Despite these successes, fundamental questions remain: what kinds of patterns can Transformers reliably learn, what are their ultimate limits, how can we train them efficiently and what mechanisms underlie their ability to generalize? One interesting direction to study these questions is in the context of learning data with pseudorandom structures. Specifically, in this paper, we use pseudo-random number generators (PRNGs) as a controlled benchmark. PRNGs are designed to pass statistical tests of randomness, yet their sequences are governed by hidden deterministic patterns. This contrast makes them an effective benchmark for testing the ultimate pattern-recognition capabilities of Transformers – in particular whether they can uncover hidden recurrence, scale to practical prediction tasks, and reveal the mechanisms that support generalization to unseen regimes.

PRNGs also comprise a fundamental primitive in cryptography. Like all primitives, their cryptographic security is based on hardness assumptions and it is therefore imperative to understand the extent to which modern AI systems can successfully attack them.

In this work, we focus on the widely used non-cryptographic family of Permuted Congruential Generators (PCGs) (O'Neill, 2014). They are practically relevant as the default generator in NumPy. PCG generates outputs based on the recurrence:

$$s_i = (a s_{i-1} + c) \bmod m, \qquad x_i = f(s_i), \tag{1}$$

where $s_i$ is the hidden LCG state at step $i$, and $x_i$ is the output. The parameters $a$, $c$, and $m$ denote the multiplier, increment, and modulus, respectively, and are fixed for a given generator. The function $f$ consists of a series of shifts, XORs, rotations and truncations to improve statistical quality and increase prediction difficulty. Transformers can learn linear congruential generators (LCGs) (Tao et al., 2025), but PCGs are far tougher: they pass BigCrush at only 49-bit state ($m=2^{49}$) or less, whereas LCGs require 88 bits ($m=2^{88}$) (O'Neill, 2014; L'Ecuyer & Simard, 2007). Our main findings are as follows:

**In-context prediction across PCG variants:** Transformers can perform in-context prediction of PCG sequences from multiple variants without explicit knowledge of the generator, and they generalize to unseen parameters $(a, c)$. This capability goes beyond classical PCG attacks (Bouillaguet et al., 2020), which assume the modulus $m$ and multiplier $a$ are known and exploit the recurrence and permutation directly. Predictions remain remarkably robust under truncation: even when only the highest bit of $s_i$ is retained in the output $x_i$, the model achieves accuracy far above random guessing.

**Scaling law with modulus:** We evaluate PCGs with modulus $m$ ranging from $2^{14}$ to $2^{22}$. The number of in-context sequence elements required to exceed 90% prediction accuracy scales as $\sqrt{m}$. This scaling is steeper than the $m^{0.25}$ law observed for LCGs.

**Curriculum is essential for large-modulus training:** At large scales ($m \geq 2^{20}$), direct training fails within the fixed budget (75k steps, batch size 512): models enter a prolonged stagnation phase with minimal loss reduction. A curriculum learning strategy is found to be essential to surmount this difficulty. The model is initialized with weights from a model trained on a smaller modulus. During training, 1% of sequences are sampled from the smaller modulus, with this probability decayed to zero over the course of training. The curriculum provides two main benefits: (1) it removes the initial loss stagnation phase and yields substantially stronger final performance under the same budget, and (2) it broadens the range of stable learning rates.

**Interpretability of learned representations:** Principal component analysis (PCA) of the embedding matrix reveals that when learning PCGs, the model spontaneously organizes tokens by features in their binary representations. In first two principal components, embeddings cluster by the number and arrangement of contiguous zero runs, a rule that remains consistent across different moduli. This structure emerges naturally when training on PCGs that apply rotations before the output, suggesting that the model has internalized the invariances inherent to the generator. When trained jointly on sequences from multiple distinct PCG variants, we show how the model's intermediate activations learn to differentiate between sequences from different variants.

## 2 EXPERIMENTAL SETTINGS

Our experiments are designed to isolate how different aspects of PRNG structure, model configuration and training strategies affect prediction performance. Here we describe the generator variants, the datasets for training and evaluation, and the model architecture and training setups.

### 2.1 PCG VARIANTS

PCGs come in different varieties, depending on the precise set of shifts, XORs, rotations and truncations, encapsulated in the function $f$ in Eq. 1. When $a$ and $c$ are chosen according to the Hull–Dobell theorem (Hull & Dobell, 1962), the state sequence $s_i$ in Eq. 1 achieves the maximal period $m$. For power-of-two moduli, however, the bits of $s_i$ exhibit position-dependent periodicities: the $k$-th least significant bit cycles with period $2^k$, far shorter than the full state period $m$ (Knuth, 1997). This makes the low-order bits especially weak, revealing structural patterns in the generator. PCG per-

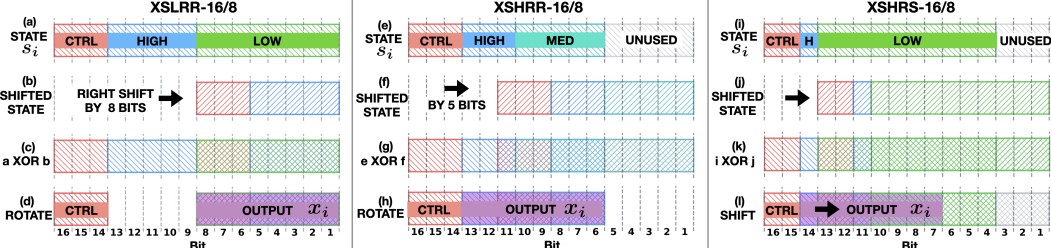

Figure 1: Depiction of PCG protocols at $m = 2^{16}$ with 8-bit output. **Left: XSLRR-16/8.** (a) State $s_i$. The top 3 bits are control bits. (b) $s_i$ is right-shifted by 8 bits. (c) The shifted state is XORed with $s_i$. (d) The lower 8 bits are retained and rotated right by the value of the control bits to produce the output. **Middle: XSHRR-16/8.** (e) State $s_i$, with the top 3 bits as control; the lowest few bits are unused. (f) $s_i$ is right-shifted by 5 bits. (g) The shifted state is XORed with $s_i$. (h) The upper 8 bits immediately following the control bits are retained and rotated right by the control bits to produce the output. **Right: XSHRS-16/8.** (i) State $s_i$, with the top 2 bits as control bits. (j) $s_i$ is right-shifted by 3 bits. (k) The shifted state is XORed with $s_i$. (l) Starting from after the control bits, the output window is right-shifted by the control bits, producing the output.

mutations mitigate this weakness by redistributing high-period structure across all bit positions using operations like XOR, shifts, and rotations. We consider the following variants:

- **TLCG** (Truncated LCG): Outputs only the high bits of the state. Part of the information of the internal state is hidden by the truncation.
- **XSLRR** (XORShift Low with Random Rotation): The state is right-shifted by half the bit length of $m$ and XORed with the original state, improving the quality of the lower half bits. This lower half is retained and rotated by an amount determined by the control bits.
- **XSHRR** (XORShift High with Random Rotation): Applies a right-shift smaller than XSLRR, then XORs with the original state. The higher bits are retained and rotated by an amount determined by control bits.
- **XSHRS** (XORShift High with Random Shift): Applies a smaller right-shift than XSLRR and XSHRR, followed by an XOR with the original state. The output window begins immediately after the control bits and is shifted right by an offset determined by those bits.

The permutations are illustrated in Figure 1. Bits are labeled from most significant (left, bit 16) to least significant (right, bit 1). Top row shows the internal state $s_i$, where the $k$-th bit in $s_i$ has period $2^k$. The lower three rows show the function $f$. Bits are split into high and low, with the low bits enhanced by the higher bits during the permutation; cross-hatched overlaps mark areas enhanced by XOR. The final rotation and shift in the permutation are controlled by the top bits of the state. This ensures that all bits in the output inherit the full period of the highest bit, which is $m$. A full description of the initial-shift calculation and pseudo-code for each generator is given in Appendix A. In practice, PCGs typically adopt a power-of-two modulus $m = 2^{\text{state size}}$, ensuring that the control bits achieve maximal period. We denote generators as *generator type-state size/output size*; for example, XSLRR-16/8 refers to an XSLRR generator with a 16-bit state and an 8-bit output.

## 2.2 DATASETS

We consider two settings:

- **Separate**: Training and test sets each contain sequences from the output of a single generator type, with no mixing between types.
- **Combined**: Training and test sets contain sequences from all four generator types.

In both cases, test sequences are generated from $a, c$ values not seen during training. The **combined** setting is more challenging, as the model must simultaneously learn and distinguish multiple generation rules, effectively forming a multi-task problem across PRNG variants. The separate setting, by contrast, isolates each variant, simplifying analysis. For scaling studies on dataset size, model size, and modulus, we focus on the XSLRR variant.

For a given modulus $m$, we select $a$ and $c$ according to the Hull–Dobell Theorem to ensure maximal period. The training set consists of sequences of length $L+1$, generated using $n_a$ distinct multipliers $a$ and $n_c$ distinct increments $c$. Each $(a, c)$ pair contributes one sequence.

Specifically: For all experiments at $m = 2^{16}$, we fix the sequence length to $L+1 = 513$. For $m \geq 2^{16}$, we increase the sequence length, setting $L > \frac{1}{2}\sqrt{m}$ to provide sufficient context. At $m = 2^{16}$ (except in dataset scaling experiments), we use $n_a = n_c = 1024$, giving a dataset of $1024 \times 1024 \times 513 \approx 5.4 \times 10^8$ tokens. At $m = 2^{22}$, we use $n_a = n_c = 2048$ and $L+1 = 1280$, giving a dataset of $2048 \times 2048 \times 1280 \approx 5.4 \times 10^9$ tokens.

## 2.3 MODEL AND TRAINING SETUP

We train Transformers to autoregressively predict the next number in sequences generated by PRNGs. Given an input $x_0, x_1, \ldots, x_{L-1}$ of length $L$, the model outputs predictions $\hat{x}_1, \hat{x}_2, \ldots, \hat{x}_L$. We use a GPT-style decoder-only Transformer (Radford et al., 2019) with Rotary Positional Embeddings (RoPE) (Su et al., 2023). Except in the model scaling experiments, models use $n_{\text{layers}} = 4$ layers, $n_{\text{heads}} = 8$ attention heads, and an embedding dimension of $d_{\text{model}} = 1024$. The vocabulary size is $2^k$ when predicting $k$-bit outputs. For example, at $m = 2^{22}$ with $k = 11$, the vocabulary size is 2048, and the model has 52M parameters. We did not use bit-wise tokenization because it increases the sequence length by a factor of $k$, making training substantially more expensive. Models are trained with cross-entropy loss and the AdamW optimizer (Loshchilov & Hutter, 2019), using a batch size of 512 for 50k–100k training steps. The learning rate uses a linear warm-up followed by cosine decay. The context length is $L$, corresponding to sequences of length $L+1$. Training details are provided in Appendix B.

## 3 TRANSFORMERS CAN IN-CONTEXT LEARN PCGS

We find that Transformers achieve reliable in-context prediction across diverse PCG variants. As shown in Figure 2(a,c), a single model trained on the **combined** dataset reaches over 90% test accuracy after having seen 512 in-context elements of a test sequence, across all PCG variants. We use "position index" to denote the location $i$ within the predicted sequence. At position $i$, the model predicts the token $\hat{x}_i$ given all previous tokens $x_{0:(i-1)}$. Training runs for 100k steps (about 8 epochs). For all generators we fix the generator state to 16 bits and the output to 8 bits. For XSLRR and XSHRR we evaluate both 2- and 3-control-bit (cb) configurations, while for XSHRS the maximum feasible number of control bits is 2, since larger values would shift the output window beyond the available state length. Transformers can simultaneously learn multiple recurrence rules, whereas classical cracking algorithms are tailored to a single generator. We observe systematic differences in convergence: truncated LCGs are learned fastest; permutations with more control bits converge more slowly and reach lower accuracy. When trained on **separate** generator datasets (Figure 2 b,d), models converge faster and achieve near-perfect accuracy after having seen 128 in-context elements of the test sequence. This confirms that each generator type is fully learnable on its own. Each model is trained for 50k steps, corresponding to 24 epochs.

The increased difficulty of the **combined** setting stems from the need to infer which permutation generated the sequence, as evidenced by the clear generator-wise separation emerging in the middle layers shown in Section 6.2. For both settings, test sets are generated from unseen $a$ and $c$ values, demonstrating generalization to unseen parameters. This is beyond current classical attacks, which require prior knowledge of both $m$ and $a$. Accuracy–position curves (Figure 2c,d) exhibit step-like improvements at powers of two. This is also observed in LCGs (Tao et al., 2025), where models exploit bit periodicity and show sudden accuracy gains once low-order bits complete their cycle in context. The persistence of this phenomenon in PCGs shows that, despite added permutations, significant residual bit-wise patterns appear at certain positions in the sequence that the model can exploit, although we have not studied their precise nature here. As shown in Appendix H.2, the model's attention patterns reflect this periodic structure.

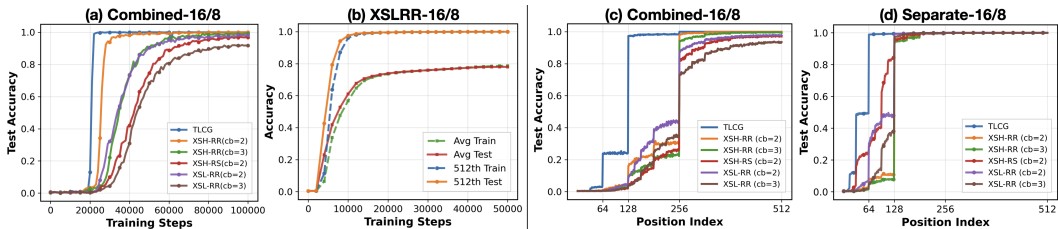

Figure 2: **(a)** Test accuracy at the 512th token during training on **combined** datasets of diverse PRNG variants. **(b)** Accuracy during training on XSLRR-16/8 dataset. "512th" refers to the model's prediction accuracy at the 512-th token. "Avg" denotes accuracy averaged across all token positions. **(c)** Final test accuracy by position index for **combined** training. **(d)** Final test accuracy when trained **separately** on each generator type, where all variants achieve near 100% accuracy with only 128 in-context elements.

# 4    WHAT LIMITS PREDICTION PERFORMANCE?

## 4.1    EFFECT OF TRUNCATIONS

To quantify the difficulty introduced by truncation, we study truncated LCGs where the low bits of the internal state are hidden and only the top $k$ bits are retained as output. For $m = 2^{16}$, this yields a $2^{16-k}$-to-1 mapping from states to outputs, so smaller $k$ increases ambiguity. To examine this effect, we train separate models for each $k$ (Figure 3, left). Despite the severe information loss, the models are surprisingly robust to truncation. Even with $k = 1$, the model attains 95% accuracy at the 256th element, far above the random-guessing baseline of $1/2^k$. At earlier positions (e.g., the 64th element), performance is lower under heavy truncation but improves quickly as $k$ increases. These results indicate that Transformers can extract patterns even from heavily truncated outputs, with longer contexts compensating for reduced information.

## 4.2    SCALING STUDIES

Practical PCGs, such as the XSLRR-128/64 generator used as NumPy's default generator, operate at a scale far beyond the 16-bit state settings. To bridge this gap, we study how performance on XSLRR changes when scaling along three axes: modulus $m$, dataset size, and model capacity.

**Effect of Generator Modulus:** We first analyze how the modulus $m$ affects prediction performance. Using a 4-layer, 8-head Transformer, we evaluate moduli ranging from $m = 2^{14}$ to $m = 2^{22}$ and observe a clear scaling law: the number of sequence elements required to reach at least 90% test accuracy grows as $\frac{1}{2}\sqrt{m}$. This relationship is shown in Figure 3 (middle, right) and indicates that context length becomes the primary bottleneck as the modulus increases. Compared to LCGs, where the requirement grows as $m^{0.25}$ (Tao et al., 2025), PCGs demand substantially longer contexts, reflecting the information obscuration introduced by truncation and permutations. If we change the accuracy threshold from 90% to $\epsilon + 1/\sqrt{m}$ or $\gamma/\sqrt{m}$, where $1/\sqrt{m}$ is the threshold for random guessing, we find the scaling law for number of required in-context sequence elements $\propto m^{\beta}$, with $\beta \in [0.4, 0.5]$ and $[0.33, 0.34]$, respectively. (See Appendix C.1).

In Appendix C.2 we compare our models' inference time compute scaling ($\propto m^{0.53}$ for $L \leq d_{\text{model}}$, which would $\propto m$ once the context length becomes significant) for achieving over 90% accuracy to a brute force search baseline ($\propto m^{2.5}$); more efficient architectures, such as state-space models (Gu & Dao, 2024) or efficient attention mechanisms, could improve the inference-time compute scaling law for ML-based attacks.

At large moduli, we use pretrained initialization combined with curriculum training (see Section 5 for details).

**Effect of Dataset Size:** We assess how much data is required to solve the XSLRR-16/8 prediction task by varying the number of distinct $(a, c)$ pairs. For $m = 2^{16}$, there are 16,384 valid multipliers $a$ and 32,768 valid increments $c$ under the Hull–Dobell conditions, yielding over $5 \times 10^8$ possible $(a, c)$ pairs. In practice, however, we find that only a small subset is sufficient to achieve generalization, with $n_a = n_c = 1024$ already providing enough diversity (Figure 4, left). Moreover, as

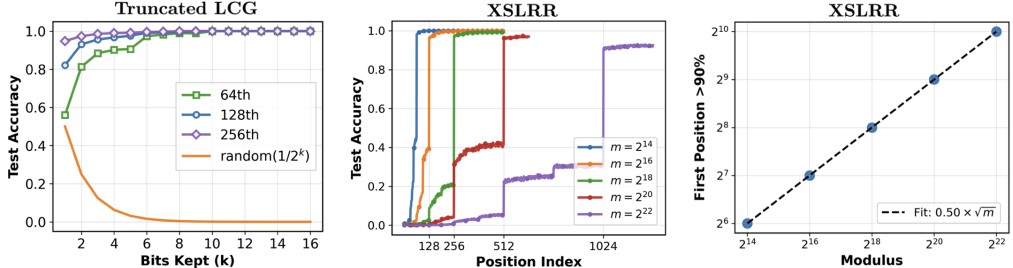

Figure 3: **Left:** Prediction accuracy at the 64th, 128th, and 256th sequence positions as a function of bits kept ($k$) in truncated LCGs with $m = 2^{16}$. Accuracy improves with larger $k$ and longer context, remaining far above the random baseline $1/2^k$ even under severe truncation. **Middle:** For XSLRR, accuracy improves stepwise as more context is observed, with reliable predictions emerging once the context length reaches exactly $0.5\sqrt{m}$ elements. **Right:** Context length required to exceed 90% test accuracy scales as $\frac{1}{2}\sqrt{m}$ with modulus $m$.

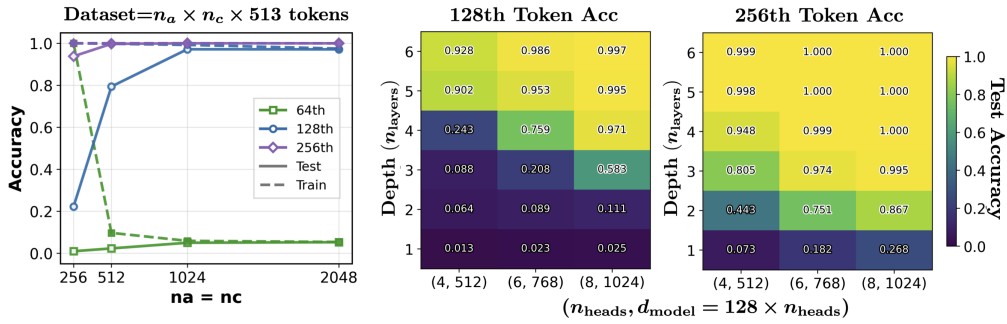

Figure 4: Scaling studies of dataset size and model capacity. **Left:** Prediction accuracy at positions 64, 128, and 256 as a function of dataset size ($n_a \times n_c$ sequences). Accuracy improves rapidly with larger datasets and saturates once sufficient diversity is reached. **Middle and Right:** Test accuracy heatmaps across model depth ($n_{\text{layers}}$) and number of heads ($n_{\text{heads}}$), evaluated at positions 128 and 256. Larger models achieve higher accuracy, with nearly perfect prediction at 128 positions once $n_{\text{layers}} \geq 4$ and $n_{\text{heads}} \geq 8$.

dataset size increases, training accuracy at early positions (e.g., the 64th) decreases while test accuracy at intermediate positions (e.g., the 128th) improves, reflecting a shift from memorization to more generalizable strategies. All experiments use a fixed budget of 50k steps with batch size 512. Appendix D shows that increasing $n_a$ or $n_c$ has equivalent benefits, with no clear advantage from expanding one parameter over the other.

**Effect of Model Size:** We evaluate performance on XSLRR-16/8 while varying Transformer depth and width. Model width is controlled through the number of attention heads while keeping head dimension fixed at 128. Figure 4 shows test accuracy at positions 128 and 256 across model sizes. Larger models need only half as many observed elements, matching smaller models' 256th-position accuracy by the 128th position, suggesting that increased scale allows the model to develop more element-efficient strategies. Appendix E shows an 1-layer model solves XSLRR 14/7, indicating that depth 1 can suffice for small-modulus PCG variants.

## 5 CURRICULUM LEARNING

Training directly on large-modulus generators leads to slow convergence and can be unsuccessful within a fixed compute budget. Here we show that curriculum learning strategies, where we incorporate data from smaller moduli, are crucial in successfully training at large moduli.

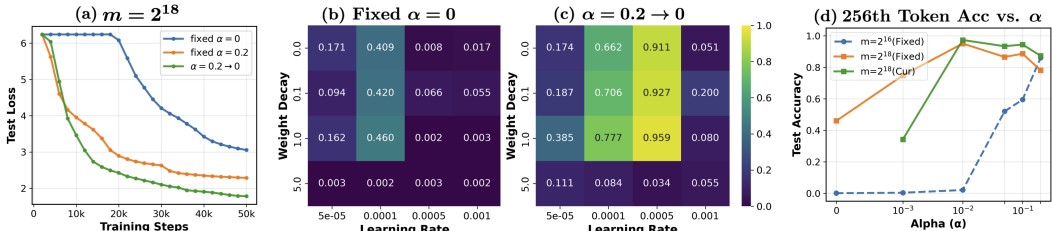

Figure 5: Effect of mixing smaller-modulus data on training stability and final accuracy. **(a)** Test loss on $m=2^{18}$ under three training setups: training only on $m=2^{18}$ (blue), fixed mixing with $\alpha=0.2$ (orange), and curriculum mixing starting at $\alpha=0.2$ and decaying to 0 over 40k steps (green). **(b,c)** Learning-rate and weight-decay landscapes for 256th-token accuracy on $m=2^{18}$, comparing training solely on $m=2^{18}$ (b) versus with the curriculum (c). **(d)** Test accuracy at the 256th token for both $m=2^{16}$ and $m=2^{18}$ under fixed mixing and curriculum mixing as the initial $\alpha$ varies.

## 5.1 DATA MIXING STRATEGIES: FIXED RATIO VS. CURRICULUM

To evaluate how mixed-modulus training improves large-modulus performance, we combine sequences from XSLRR-18/9 ($m = 2^{18}$) with additional examples from XSLRR-16/8 ($m = 2^{16}$), where the mixing ratio $\alpha$ specifies the probability of sampling from the $m = 2^{16}$ dataset. In the curriculum setting, we decay $\alpha$ to zero over 40k steps, whereas in fixed-$\alpha$ training $\alpha$ is held constant. As shown in Figure 5(a), both approaches remove the long stagnation observed when training solely on $m = 2^{18}$. Figure 5(b,c) compare the learning-rate/weight-decay landscapes with and without curriculum. Curriculum training substantially broadens the range of stable learning rates, enabling much larger step sizes without instability. Figure 5(d) shows how varying the initial mixing ratio $\alpha$ influences prediction accuracy under both fixed and curriculum training. The blue and orange curves show test accuracy on $m=2^{16}$ and $m=2^{18}$ respectively under fixed-$\alpha$ training. The green curve shows test accuracy on $m=2^{18}$ under curriculum training. These results demonstrate two key effects: (1) Even when $m=2^{16}$ itself is not learned, mixing a small fraction of its data substantially boosts performance on $m=2^{18}$; (2) As $\alpha$ increases, the model learns to handle both moduli simultaneously. Curriculum consistently yields higher accuracy on $m=2^{18}$ than fixed-$\alpha$, achieving its best performance at initial mixing ratio $\alpha = 1\%$. In Appendix F, we compare cosine, exponential, linear, and step decay schedules for $\alpha$ and find exponential decay yields the best performance.

## 5.2 PRETRAINED INITIALIZATION: LEVERAGING SMALLER-MODULUS MODELS

Using a model trained on a smaller modulus at initialization can be viewed as a discrete form of curriculum learning. Instead of gradually transitioning from easy to hard data, the model is first exposed to the smaller modulus and then fine-tuned on the harder task. To evaluate whether the recurrence and permutation structures learned at smaller moduli transfer effectively to larger ones, we train models on XSLRR-20/10 ($m_{\text{test}}=2^{20}$) and compare pretrained initialization against random initialization (Figure 6a,b). We evaluate four settings: (1) random initialization: train directly on XSLRR-20/10 from scratch; (2) pretrained initialization: initialize from a model trained on XSLRR-18/9 ($m=2^{18}$), then train on XSLRR-20/10; (3) smooth curriculum: start from random initialization but mix in data from XSLRR-16/8 during training (as Figure 12 shows, mixing in XSLRR-18/9 gives little benefit); (4) smooth curriculum + pretrained initialization: initialize from XSLRR-18/9 model and mix in XSLRR-18/9 data. For curriculum training, the probability of sampling from the smaller-modulus dataset starts at $\alpha = 0.01$ and decays exponentially to zero over the first 50k steps, after which training continues for an additional 25k steps exclusively on the target modulus. For pretrained initialization, the overlapping portion of the embedding matrix is transferred, while additional tokens required for the larger vocabulary are randomly initialized. Although higher moduli require longer contexts, RoPE's extended positional scaling allows pretrained models to adapt to the larger sequence lengths.

As shown in Figure 6(a,b), training from random initialization without curriculum does not converge within the allotted 75k steps, remaining stuck at high loss and only 4% accuracy at the 640th token. Pretrained initialization provides the main benefit: models skip the long stagnation phase, converge faster, and consistently reach higher final accuracy than those trained from random initialization. Curriculum provides additional gains when combined with pretraining and, when used alone, par-

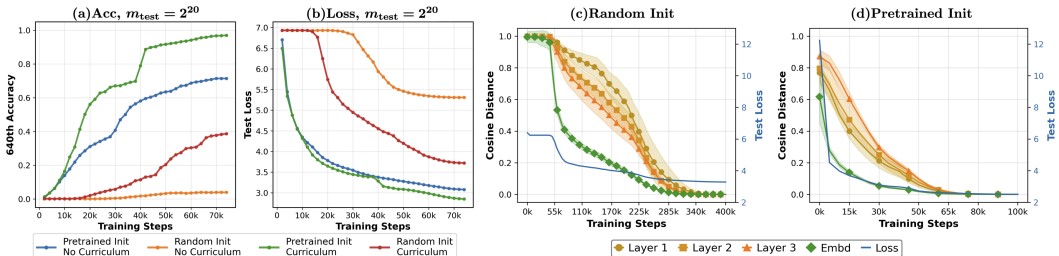

Figure 6: Impact of pretrained initialization and curriculum training on scaling to larger moduli. **(a)** Test accuracy at the 640-th token on $m_{\text{test}}=2^{20}$ across training steps. **(b)** Test loss on $m_{\text{test}}=2^{20}$ across training steps. **(c)** Cosine distance of parameters from their final values and test loss when training a 3-layer Transformer from scratch for 400k steps. **(d)** Cosine distance of parameters from their final values and test loss when training the same model for 100k steps starting from a pretrained model. Shading in (c) and (d) shows the standard deviation

tially mitigates stagnation. Together, these results show that both pretraining and curriculum are crucial for scaling to larger moduli under fixed training budgets.

## 5.3 PRETRAINING AS A SHORTCUT TO STABLE REPRESENTATIONS

To understand why pretrained initialization accelerates training, we examine how model weights evolve over time. We train a three-layer Transformer on XSLRR-18/9 under two settings: (i) from scratch for 400k steps, and (ii) for 100k steps starting from a pretrained model on XSLRR-16/8. Figure 6(c,d) tracks how each layer of the model changes during training by measuring the cosine distance between parameters at each step and their final trained values. The green curve represents the token embedding matrix, the orange curves correspond to the three Transformer layers, and the blue curve (right axis) shows test loss. In both training regimes, the embedding layer reaches a usable representation first, after which the deeper layers evolve more rapidly. With pretrained initialization the model starts much closer to its final state: embeddings reach this usable representation almost immediately, allowing deeper layers to adapt right away. This strong embedding space prior shortens stagnation and accelerates convergence. As shown in the next section and in Appendix H.1, a clear structure in the embedding space persists across moduli, supporting this transfer effect.

## 6 INTERPRETABILITY OF MODEL REPRESENTATIONS

### 6.1 TOKEN EMBEDDINGS

To understand how Transformers model PCG patterns, we analyze the token embedding layer of a model trained on XSLRR-16/8. We apply principal component analysis (PCA) to the embedding matrix. The top principal components align with bit-level statistics of the tokens, reflecting the symmetries of the generator. The first two components in Figure 7 form clear clusters. To formalize the structure, we use *zero-run notation* $Z(a_1, a_2, \ldots, a_k)$, where each $a_i$ denotes the length of a contiguous run of `0`s between `1`s. The zero-run patterns and representative binary tokens for each cluster are shown in Figure 7(Right), with the complete listing provided in Table 1. We find that the first principal component (PC1) perfectly correlates the total number of zero bits $N_0$ in a token, while the second (PC2) perfectly correlates with the number of zero runs. Vertical bands in Figure 7 correspond to constant $N_0$ (e.g., clusters 6, 12, 16, and 18 all have $N_0 = 4$), while horizontal groupings reflect constant run counts (e.g., clusters 10–14 all contain two zero runs). PC3 captures an even–odd bit imbalance in the token representation: $\text{PC}_3(x) \propto (b_0 + b_2 + b_4 + b_6) - (b_1 + b_3 + b_5 + b_7)$, where $b_i$ denotes the $i$-th bit of the token. An analysis of higher components is provided in Appendix H.1.1. As shown in Appendix H.1, these grouping rules persist at larger moduli. The persistence of this learned structure across moduli explains why pretrained initialization is so effective.

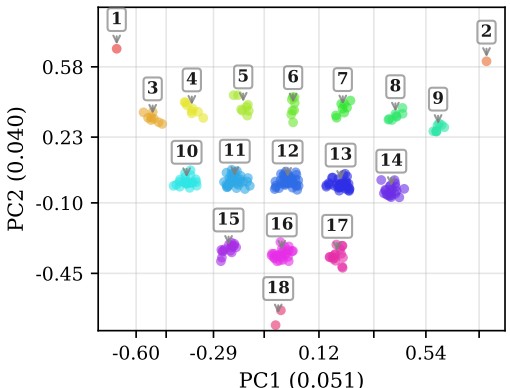

| Cluster | Zero-run pattern | Example |
|---------|-----------------|---------|
| 1 | All 1s | 11111111 |
| 2 | All 0s | 00000000 |
| 3 | Z(1) | 01111111 |
| 4 | Z(2) | 00111111 |
| 5 | Z(3) | 00011111 |
| 6 | Z(4) | 00001111 |
| 7 | Z(5) | 00000111 |
| 8 | Z(6) | 00000011 |
| 9 | Z(7) | 00000001 |
| 10 | Z(1,1) | 01011111 |
| 11 | Z(2,1) | 00101111 |
| 12 | Z(3,1), Z(2,2) | 00010111 |
| 13 | Z(4,1), Z(3,2) | 00001011 |
| 14 | Z(5,1), Z(4,2), Z(3,3) | 00000101 |
| 15 | Z(1,1,1) | 01010111 |
| 16 | Z(2,1,1) | 00101011 |
| 17 | Z(3,1,1), Z(2,2,1) | 00010101 |
| 18 | Z(1,1,1,1) | 01010101 |

Figure 7: PCA of token–embedding matrix for XSLRR-16/8 (left) and cluster summary (right). Tokens group by rotation-invariant zero-run structures; full table with all tokens provided in appendix.

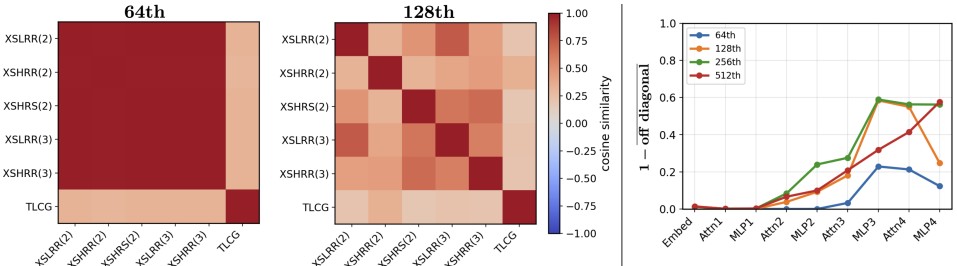

Figure 8: Cosine similarity of representations across different PRNG variants for a 4-layer Transformer trained on the **combined** dataset. Numbers in parentheses indicate the number of control bits. **Left**: At the 64th token position, third-layer MLP outputs already separate truncated LCGs from PCG variants, though PCG types remain highly overlapping. **Middle**: At the 128th token position, the same MLP outputs cleanly separate all PCG variants. Variants with the same permutation type but different control-bit counts are more similar to each other than to other types. **Right**: Generator separation across the network for selected token positions (64th, 128th, 256th, 512th) defined as $1 -$ mean off-diagonal cosine similarity. Higher values indicate stronger generator separation.

## 6.2 GENERATOR SEPARATION

When trained on **combined** datasets, the model develops a permutation-agnostic grouping of tokens(Figure 27). This raises the question of how the model is able to predict different PRNG variants at test time. Despite receiving no explicit supervision about generator identity, the model's internal representations spontaneously distinguish PRNG variants. In a 4-layer model, this structure emerges most clearly in the MLP output of the third Transformer block: by the 64th token position, the model already distinguishes truncated LCGs from PCG variants, and by the 128th token, it cleanly differentiates between all PCG variants (see Figure 8, left and middle). To quantify this effect across layers, we plot the average off-diagonal cosine dissimilarity between generators at each position (Figure 8 right). Separation is weakest in the embeddings and first layer, rising sharply through the middle MLP and attention layers, suggesting that model first forms a shared representation of the underlying recurrence and then, in deeper layers, refines generator-specific distinctions. In Appendix H.3, we present head-level ablations showing that several attention heads in the last three layers specialize differently across generator variants.

## 7 RELATED WORK

**Cracking PCGs:** Classical approaches to cracking PRNGs rely on exploiting algebraic structure with strong assumptions about the generator. Bouillaguet et al. (2020) present an attack on XSLRR-128/64 that assumes knowledge of the multiplier, modulus, and permutation. The internal state can

be recovered from 64 outputs via a guess-and-procedure. An asymptotic scaling law with modulus $m$ is not provided in those attacks, although the wall-clock time is significant ($20,000$ CPU hours in the worst case). In our work, the models must discover the hidden structure from training data alone, learning to predict outputs without explicit knowledge of the recurrence or transformation rules.

**Curriculum Learning:** Many studies have explored curriculum learning (Bengio et al., 2009). Wu et al. (2021) finds that on standard benchmarks, explicit curricula offer little advantage when training steps are sufficient, but can yield higher accuracy and stability under limited compute or noisy data. Garg et al. (2023) observes that curriculum training can speed up training drastically when training Transformers to in-context learn linear functions. Recently Saxena et al. (2024) observed that curriculum learning strategies can be helpful in modular addition.

**AI for Cryptography**: There is a classic duality between machine learning and cryptography (Rivest, 1991). Recently there has been increased interest in using modern AI systems to attack cryptographic schemes, such as the learning with errors problem (Wenger et al., 2022). Our work builds on Tao et al. (2025), which uses transformers to learn vanilla LCGs.

**Interpretability and Modular arithmetic:** A growing body of work examines how Transformers learn modular arithmetic tasks, uncovering phenomena such as grokking and structured internal representations (Power et al., 2022; Gromov, 2023; Zhong et al., 2023; Nanda et al., 2023; Doshi et al., 2024; Charton & Kempe, 2024). Prior studies (Liu et al., 2022; He et al., 2024) also find emergent structures in embedding matrices and interpretable attention patterns. Our work extends this literature by studying modular arithmetic tasks involving *permutation structures* and identifying a novel pattern in embedding space.

**Hidden Markov Models (HMMs):** Prior in-context learning studies on HMMs treat the transition and emission as arbitrary probability tables without internal structure, which constrains these tasks to very small state spaces (typically $\leq 64$) and requires observing all transition and emission examples (Wei et al., 2022; Xie et al., 2022; Dai et al., 2025). A PCG can be written in HMM form, but its transition and emission are deterministic algebraic operations rather than stochastic tables. This difference enables Transformers to learn the underlying update rule and generalize to unseen parameters $(a, c)$, whereas HMM-based ICL tasks require observing each transition instance.

## 8 DISCUSSION

This work uses PRNG prediction as a controlled setting to probe the pattern recognition capabilities and limitations of Transformers. Our results show that the model can learn several PRNG variants, generalize to unseen parameters $(a, c)$, going beyond classical cracking methods that require explicit knowledge of generator parameters. At the same time, scaling to larger moduli reveals clear constraints: successful generalization requires curriculum learning and sufficient context length, with the required context empirically scaling as $\mathcal{O}(\sqrt{m})$. Our PCA identifies bit-level structure in the embedding space, shedding light on how the model represents outputs by exploiting the symmetries underlying the PCG algorithm. A full mechanistic account of the solution implemented by the model, and an exploration of whether Transformers can approach cryptographically secure generators are important directions for future work.

## ACKNOWLEDGMENTS

We thank Dayal Singh Kalra, Tianyu He, and Darshil Doshi for their valuable discussions and feedback and for collaborations on related prior work. This work is supported by NSF DMR-2345644 and by the Simons Collaboration on Physics of Learning and Neural Computation, which is a grant from the Simons Foundation (SFI-MPS-POL-00012574-09). We acknowledge the University of Maryland High Performance Computing Cluster for providing the computational resources used in this study.

REPRODUCIBILITY STATEMENT

All model architectures, training hyperparameters and data generation procedures are full described in Appendix B. We release our implementation at `https://github.com/TaoT1998/learn-pcg`.

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

# A    PERMUTED CONGRUENTIAL GENERATORS

In this section we review the background of LCGs, truncated LCGs and PCGs.

## A.1    HULL–DOBELL THEOREM.

For an LCG

$$s_i = as_{i-1} + c \pmod{m}, \tag{2}$$

the sequence $\{s_i\}$ has full period $m$ if and only if the following three conditions hold:

1. $c$ and $m$ are relatively prime,

2. $a - 1$ is divisible by all prime factors of $m$,

3. if $m$ is divisible by $4$, then $a - 1$ is also divisible by $4$.

## A.2    PERIOD OF LOW-ORDER BITS IN LCGS

Consider the LCG

$$x_{t+1} = (ax_t + c) \bmod m, \tag{3}$$

with $m = 2^K$, $c$ coprime to $m$, and $a - 1$ divisible by $4$, so that $\{x_t\}$ has full period $m$ by the Hull–Dobell theorem. Let

$$z_{t,k} = x_t \bmod 2^k \tag{4}$$

denote the lowest $k$ bits of $x_t$. Then

$$z_{t+1,k} = (az_{t,k} + c) \bmod 2^k, \tag{5}$$

so $\{z_{t,k}\}$ itself is an LCG with modulus $2^k$. Since $c$ is coprime to $2^k$ and $a - 1$ is divisible by $4$, this reduced generator achieves full period $2^k$. Thus, the $k$-th lowest bit of an LCG with power-of-two modulus cycles with period exactly $2^k$, much shorter than the full state period $m$.

## A.3    PRNG VARIANTS.

We consider three widely used PCG permutations, each defined for a $2n$-bit state with $cb$ control bits and an $n$-bit output. The internal state evolves as:

$$s_i = as_{i-1} + c \pmod{m}, \quad m = 2^{2n}. \tag{6}$$

- **XSLRR (Xorshift Low, Random Rotation).** First apply a right shift of $n$ bits and XOR with the original state, folding the high and low halves together. The low $n$ bits of the result are then retained and rotated right by the control value to produce the output. Formally:

$$\text{control bits  value: } v = s_i \gg (2n - cb), \tag{7}$$
$$\text{state XOR shifted state: } s_i' = s_i \oplus (s_i \gg n), \tag{8}$$
$$n\text{-bit output: } x_i = \text{rot}_v(s_i' \bmod 2^n), \tag{9}$$

  where $s_i \gg n$ denotes right shift $s_i$ by $n$ bits and $\text{rot}_v$ denotes an $n$-bit right rotation by $v$.

- **XSH-RR (Xorshift High, Random Rotation).** First apply a right shift by $\lfloor (n+cb)/2 \rfloor$ bits and XOR the result with the original state. The $n$ bits immediately following the control bits are then retained and rotated right by the control value to produce the output. Formally:

$$\text{control bits  value: } v = s_i \gg (2n - cb), \tag{10}$$
$$\text{state XOR shifted state: } s_i' = s_i \oplus (s_i \gg \lfloor (n + cb)/2 \rfloor), \tag{11}$$
$$n\text{-bit output: } x_i = \text{rot}_v((s_i' \gg (n - cb)) \bmod 2^n) \tag{12}$$

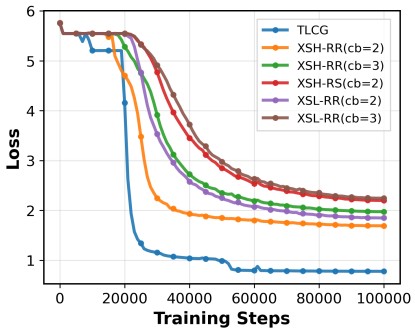

Figure 9: Test loss curves for each generator type when the model is trained on the combined dataset.

- **XSH-RS (Xorshift High, Random Shift).** First apply a right shift by $(n-cb-2^{cb}+1)$ bits and XOR the result with the original state. The $n$-bit output window begins immediately after the control bits, but its starting position is shifted further right by the control value $v$, selecting a different $n$-bit segment of the state. Formally:

$$\text{control bits  value: } v = s_i \gg (2n - cb), \tag{13}$$

$$\text{state XOR shifted state: } s_i' = s_i \oplus (s_i \gg (n - cb - 2^{cb} + 1)), \tag{14}$$

$$n\text{-bit output: } x_i = (s_i' \gg (n + v)) \bmod 2^n. \tag{15}$$

We also consider truncated LCGs, where the output is formed by retaining only the top $k$ bits of the internal state $s_i$, hiding the lower-order bits. This preserves the recurrence structure and full period of the LCG while exposing only partial information about the state. Formally:

$$x_i = s_i \gg (2n - k) \bmod 2^k, \tag{16}$$

## B  TRAINING DETAILS

**Combined Dataset (Figure 2a,c).**   For each generator type, we select $n_a=n_c=1024$ training multipliers $a$ and increments $c$ using the Hull–Dobell theorem. One sequence per $(a, c)$ pair is generated with its initial state $x_0$ sampled randomly using NumPy's RNG. All sequences from all generator types are merged into a single dataset and reshuffled at the start of each epoch to randomize the sampling order. Test loss and accuracy in Figure 2(a) and Figure 9 are computed on a held-out set with $n_{\text{test\_a}}=128$ multipliers and $n_{\text{test\_c}}=16$ increments; in Figure 2(c), evaluation uses $n_{\text{test\_a}}=128$ and $n_{\text{test\_c}}=64$.

We train a Transformer with depth $n_{\text{layers}} = 4$, $n_{\text{heads}} = 8$ attention heads, and $d_{\text{model}}=1024$ for 100k steps with batch size 512 (about 8 epochs). The learning rate is 0.0001 with weight decay 1.0, using 5000 warm up steps (linear) followed by cosine decay. Training is performed on two NVIDIA A100 GPUs and takes roughly 8 hours.

**Separate Datasets (Figure 2b,d).**   We follow the same procedure for selecting $a$ and $c$ but train a separate model on each generator type individually. Each model is trained for 50k steps with batch size 512 (roughly 4 hours on two A100 GPUs). We perform a grid search over learning rate and weight decay for each generator to ensure fair comparison across configurations.

**Truncated LCG (Figure 3 left).**   We evaluate truncated LCGs with $m = 2^{16}$, varying the number of retained bits $k$ from 1 to 16, which determines the effective output range. Each integer is tokenized into two base-256 digits, except for special cases where a smaller base performed better: for $k=7$ we use base-128 with one digit, and for $k=9$ we use base-64 because training with base-256 consistently yielded worse performance. This tokenization dramatically reduces the vocabulary size (e.g., $k=16$ would otherwise require 65,536 symbols) and empirically improves convergence. Because each number is split into two tokens, the context length doubles, and a prediction is counted correct only when both digits are predicted correctly.

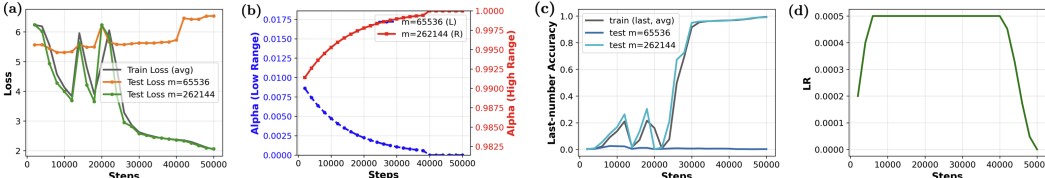

Figure 10: **Curriculum training with random initialization from** $m=2^{16}$ **to** $m=2^{18}$**. (a)** Training and test loss over steps. **(b)** Evolution of mixing ratio $\alpha$. **(c)** Last-token accuracy over time for both moduli. **(d)** Learning rate schedule during training.

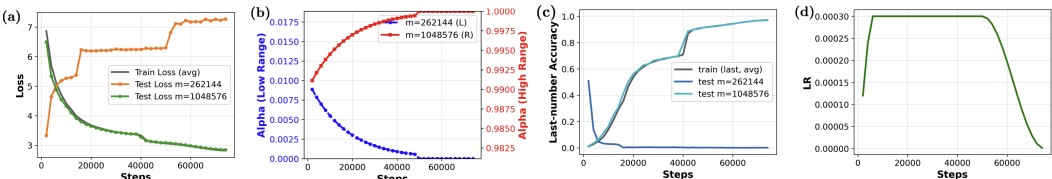

Figure 11: **Curriculum training with pre-trained initialization from** $m=2^{18}$ **to** $m=2^{20}$**. (a)** Training and test loss over steps. **(b)** Evolution of mixing ratio $\alpha$. **(c)** Last-token accuracy over time for both moduli. **(d)** Learning rate schedule during training.

Base-256 tokenization is a practical trade-off between context length and vocabulary size. With bitwise tokenization, an 8-bit output would expand to 8 tokens, increasing the context length by a factor of 8 and, leading to roughly $64\times$ more attention memory and compute. For each configuration, we train for 50k steps with batch size 512, taking roughly four hours on two NVIDIA H100 GPUs for two-digit experiments and about two hours for one-digit experiments.

**Larger Modulus (Figure 3 right, middle).** For $m = 2^{18}$, we trained a 4-layer Transformer for 50k steps (batch size 512) with context length 512 on two NVIDIA A100 GPUs (4 hours). The curriculum began with sequences from $m=2^{16}$ mixed into $m=2^{18}$ at $\alpha$=0.01 and decayed exponentially to zero over the first 40k steps. Training continued for the remaining 10k steps entirely on $m=2^{18}$. Training sets for each modulus were generated with $n_a=n_c=1024$. Using learning rate 0.0005 and weight decay 1.0, the model achieved a test accuracy on $m=2^{18}$ of 0.9907 at position 512. Figure 10 shows the curriculum, learning rate and learning curve. For $m = 2^{20}$, we trained a 4-layer Transformer for 75k steps (batch size 512) with context length 640 on two NVIDIA A100 GPUs, totaling roughly 10 hours of compute. Training sets for each modulus were generated with $n_a=n_c=2048$. The model was initialized from the above checkpoint trained on $m=2^{18}$ for 50k steps. The curriculum began with sequences from $m=2^{18}$ mixed into $m=2^{20}$ with sampling possibility $\alpha$=0.01, then exponentially decayed this proportion to zero over the first 50k steps. The final 25k steps were trained entirely on $m=2^{20}$. Using learning rate 0.0003 and weight decay 0.1, the model achieved a test accuracy on $m=2^{20}$ of 0.9697 at position 640. Figure 11 summarizes the curriculum experiment using pretrained initialization, transferring from $m=2^{18}$ to $m=2^{20}$. Figure 12 shows the same curriculum schedule but starting from random initialization.

For $m = 2^{22}$, we trained a 4-layer Transformer for 100k steps (batch size 512) with context length 1279 on two NVIDIA A100 GPUs, totaling roughly 12 hours of compute. The model was initialized from the previously trained checkpoint on $m=2^{20}$. Training sets for each modulus were generated with $n_a=n_c=2048$. The curriculum began with sequences from $m=2^{18}$, $m=2^{20}$, and $m=2^{22}$ mixed at initial proportions $\alpha$=0.01, 0.01, 0.98 and decayed to $\alpha$=0.0, 0.0, 1.0 over the first 75k steps. The final 25k steps were trained exclusively on $m=2^{22}$. Using learning rate 0.0005 and weight decay 0.1, the model achieved a test accuracy on $m=2^{22}$ of 0.9257 at position 1279. Figure 13 shows the curriculum and learning curves.

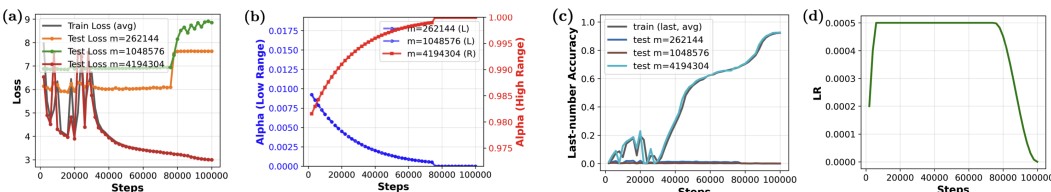

Figure 12: **Curriculum training with random initialization from** $m=2^{18}$ **to** $m=2^{20}$**.** (a) Training and test loss over steps. (b) Evolution of mixing ratio $\alpha$. (c) Accuracy at the final prediction position. (d) Learning rate schedule.

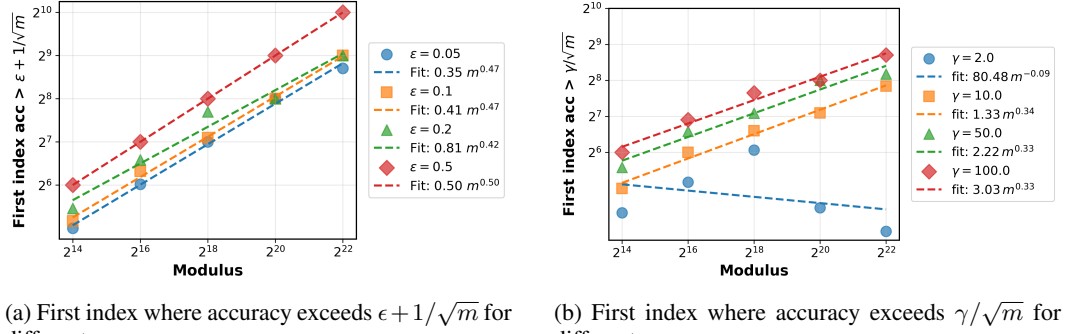

Figure 13: **Curriculum training with pre-trained initialization from** $m=2^{20}$ **to** $m=2^{22}$**.** (a) Training and test loss over steps. (b) Evolution of mixing ratio $\alpha$. (c) Last-token accuracy over time for both moduli. (d) Learning rate schedule during training.

## C  SCALING WITH GENERATOR MODULUS

### C.1  ACCURACY THRESHOLD

In the main text, we fixed the accuracy threshold at 90% to study how the required context length scales with $m$. Here, we extend this analysis to additional criteria: (1) performance exceeding random guessing by a margin $\epsilon$, and (2) a multiplicative threshold $\gamma \times$ random-guess accuracy. Random guessing corresponds to an accuracy of $1/\sqrt{m}$ for our task. As shown in Figure 14, for the additive threshold criterion ($\epsilon$ above random guessing), the fitted exponents are near $0.5$, while for the multiplicative threshold criterion ($\gamma \times$ random guessing), the exponents cluster are around $0.33$, indicating a slower scaling requirement under the multiplicative rule.

(a) First index where accuracy exceeds $\epsilon + 1/\sqrt{m}$ for different $\epsilon$.

(b) First index where accuracy exceeds $\gamma/\sqrt{m}$ for different $\gamma$.

Figure 14: Scaling of required context length with $m$ under alternative accuracy thresholds.

## C.2 COMPUTE SCALING

**Per-sequence Inference FLOPs.** Given $L$ (sequence length), $d_{model}$ (embedding dimension), $n_{layer}$ (number of layers), and $|V|$ (vocabulary size), the total inference FLOPs can be decomposed as:

$$\text{FLOPs}_{\text{attn}} = \left(4\,L\,d_{model}^2 + 2\,L^2\,d_{model}\right) n_{layer},$$

$$\text{FLOPs}_{\text{MLP}} = 8\,L\,d_{model}^2\,n_{layer} \quad \text{(MLP ratio = 4)},$$

$$\text{FLOPs}_{\text{LayerNorm}} = 2\,L\,d_{model}\,n_{layer} \quad \text{(two norms per layer)},$$

$$\text{FLOPs}_{\text{LM head}} = L\,d_{model}\,|V|,$$

$$\text{FLOPs}_{\text{Embed}} = 0 \quad \text{(input embedding lookup only; memory fetch)}.$$

So the total per-sequence inference cost is:

$$\text{FLOPs}_{\text{infer}} = \text{FLOPs}_{\text{attn}} + \text{FLOPs}_{\text{MLP}} + \text{FLOPs}_{\text{LayerNorm}} + \text{FLOPs}_{\text{LM head}} + \text{FLOPs}_{\text{Embed}}, \quad (17)$$

$$= \left(12\,L\,d_{model}^2 + 2\,L^2 d_{model} + 2\,L\,d_{model}\right)n_{layer} + L\,d_{model}|V|. \quad (18)$$

**Total Training FLOPs.** Following the standard approximation that backward $= 2\times$ forward, the total training FLOPs are:

$$\text{FLOPs}_{\text{train}} \approx 3 \times \text{BatchSize} \times \text{TrainingSteps} \times \text{FLOPs}_{\text{infer}},$$

where the factor of 3 accounts for forward pass, backward pass.

**Brute Force Baseline** We estimate a computational baseline for brute-forcing all possible combinations of multiplier $a$, increment $c$, and seed $s_0$ for the XSLRR generator.

First we estimate the compute required for generating one output from XSLRR recurrence can be estimated as: each step of the LCG recurrence requires roughly two integer operations (multiply and add), while bitwise shifts, XORs, and rotations in XSLRR are at least four operations per output. Modulo reduction is assumed free for powers of two. Thus, each output costs at least six integer operations in total.

Then, we estimate the minimum number of outputs needed to determine whether a candidate triplet $(a, c, s_0)$ is correct. Using the information theory lower bound, we compare the total number of unknown bits (from $(a, c, s_0)$) to the information conveyed per observed output, giving the smallest sample size required to uniquely identify the generator parameters. There are $m/4$ possible $a$ values and $m/2$ possible $c$ values valid under the Hull–Dobell conditions. Since there are m possible states, observing a single output from $\sqrt{m}$ possible values reduces the candidate space from m to roughly $\sqrt{m}$ possible states. The total unknown information seeing an output $x$ is

$$\log_2(m/4) + \log_2(m/2) + \log_2(\sqrt{m}) \text{ bits.} \quad (19)$$

Given that each observed output reveals $\log_2 \sqrt{m}$ bits, the minimum number of outputs required by the information lower bound is

$$\frac{\log_2(m/4) + \log_2(m/2) + \log_2(\sqrt{m})}{\log_2 \sqrt{m}} \approx 5. \quad (20)$$

Multiplying by the per-output operation cost and the size of search space yields the total operation baseline:

$$\text{OPs}_{\text{brute}} \approx \frac{m}{4} \times \frac{m}{2} \times \sqrt{m} \times 5 \times 6, \quad (21)$$

$$\approx 3.75 m^{2.5} \quad (22)$$

We assume that each integer operation has the same cost as one FLOP. Under this assumption, we compare our brute-force baseline to the measured inference and training costs of our Transformer models in Figure 15. The inference compute is dominated by the $12\,d_{model}^2 L$ term in Equation (18) because in our experiments $L \leq d_{model}$. As $L$ increases, the quadratic term $2\,L^2 d_{model}$ will eventually dominate, and the compute will scale proportionally with $m$.

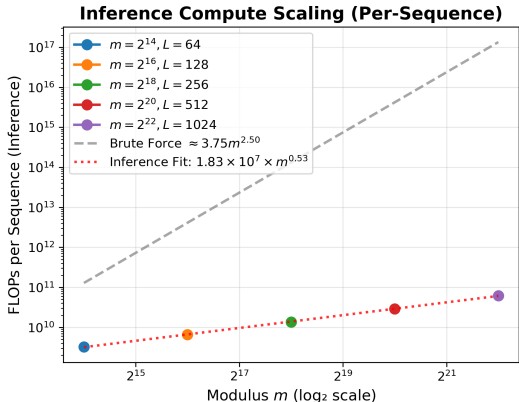

Figure 15: **FLOPs scaling with modulus.** Inference compute per sequence for models trained on PCGs with moduli $m = 2^{14}$–$2^{22}$ (context lengths required reach 90% accuracy listed in legend), compared to the brute-force baseline (dashed line). Inference cost grows far more slowly than the brute-force bound, showing the compute efficiency of Transformer-based prediction relative to direct state-space search.

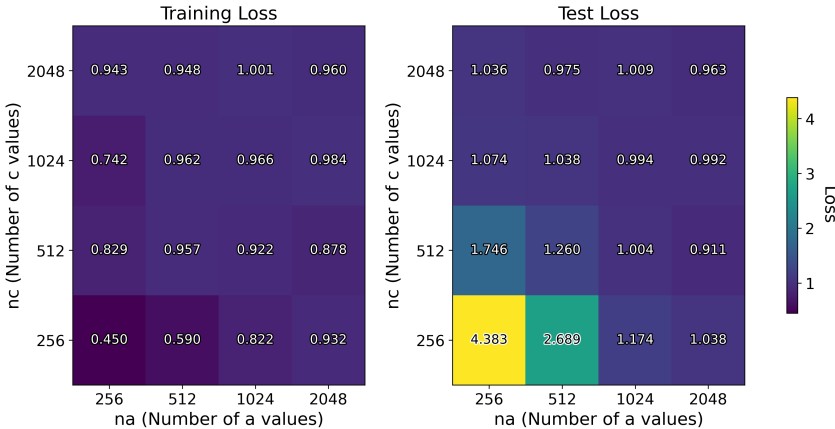

Figure 16: Training (left) and test (right) loss as a function of $n_a$ (number of multipliers $a$) and $n_c$ (number of increments $c$) for XSLRR-16/8 with 3 control bits.

# D  DATASET SIZE

We systematically varied the number of multipliers ($n_a$) and increments ($n_c$) to evaluate training-set effects. As shown in Figure 16, performance depends only on the total scale of $(n_a, n_c)$: increasing $n_a$ produces the same gains as increasing $n_c$. Moderate values ($n_a \times n_c = 512 \times 1024$) already yield low training and test loss, with little improvement beyond this point. All models are trained for 50k steps with a batch size of 512. The number of epochs ranges from about 390 for the smallest dataset ($n_a = n_c = 256$) to about 6 for the largest dataset ($n_a = n_c = 2048$). As shown in Figure 17, when trained directly on $m = 2^{18}$ without curriculum, the model shows no signs of overfitting except at the smallest setting ($n_a = n_c = 512$)—training and test losses remain closely matched across all $(n_a, n_c)$ configurations. However, unlike the smoother patterns observed for XSLRR-16/8, the heatmap here is less regular, reflecting the greater training instability at the larger modulus.

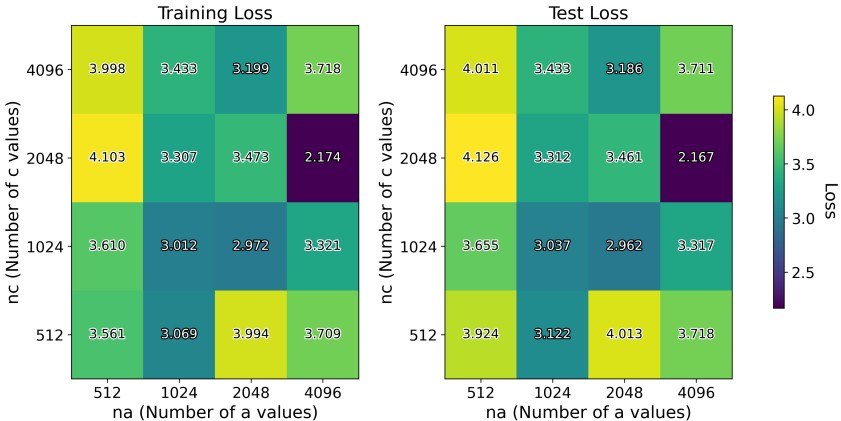

Figure 17: Training (left) and test (right) loss for XSLRR-18/9 with $m=2^{18}$ and 3 control bits, as a function of the number of multipliers $n_a$ and increments $n_c$.

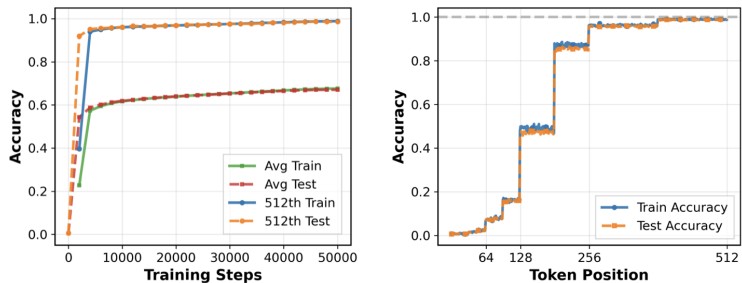

Figure 18: 1-layer, 8-head Transformer trained on XSLRR 14/7. **Left:** Learning curve. **Right:** Final performance.

# E  MODEL SIZE

## E.1  SEPARATE (XSLRR)

**Minimal Depth**  Figure 18 shows the training curve and final performance of a one-layer Transformer trained on XSLRR 14/7. The model rapidly converges and achieves 98.6% test accuracy at the 512th token, showing that depth = 1 can suffice for small-modulus PCG variants.

**Scaling**  In the main text we reported token-accuracy heatmaps for the 128th and 256th positions. Figure 19 complements this with earlier (64th) and later (512th) prediction positions. The results show that even a 1-layer model achieves above-random(44.2%) test accuracy by the 512th token, while a 6-layer model can surpass random guessing and reach 22.8% test accuracy by the 64th token. This illustrates how additional depth accelerates the emergence of useful recurrence representations at earlier positions. Figure 19 (right) plots test loss versus model size for different depth–width configurations. We observe a sharp decrease in loss as the number of parameters increases, followed by diminishing returns once the model exceeds roughly 40–70M parameters. At similar parameter counts, deeper models achieve much lower loss than simply adding heads. For example, a shallow wide model (e.g., h8d1) has far higher loss than a moderately deep one with fewer params (e.g., h4d4).

## E.2  COMBINED

Figure 20 shows the learning dynamics and final test performance of a 2-layer Transformer trained on the combined dataset. TLCG converges fastest, followed by XSHRR with 2 control bits, while

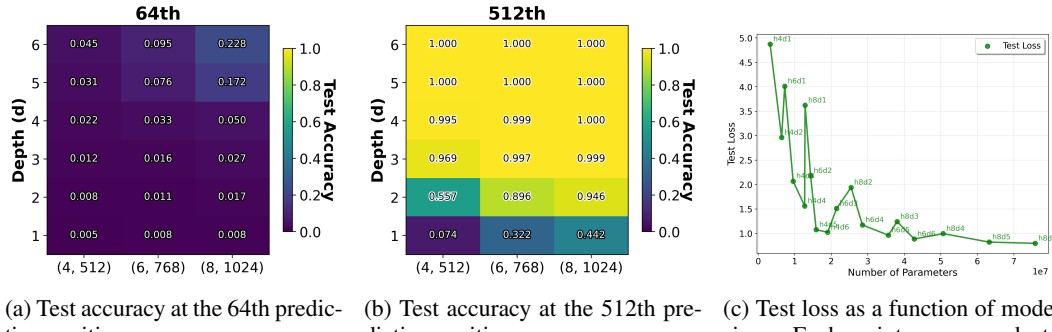

(a) Test accuracy at the 64th predic-
tion position.

(b) Test accuracy at the 512th pre-
diction position.

(c) Test loss as a function of model
size. Each point corresponds to
$(h = n_{\text{heads}}, d = n_{\text{layers}})$ heads and
depth.

Figure 19: **Effect of model depth/width and size on test accuracy. Left and Middle:** Token
accuracy heatmaps at the 64th and 512th positions. The y-axis (Depth $d$) indicates the number of
Transformer layers $n_{\text{layer}}$, and the x-axis shows model size in terms of attention heads and embedding
dimension $(n_{\text{heads}}, d_{\text{model}})$. **Right:** Test loss versus model size (number of parameters)

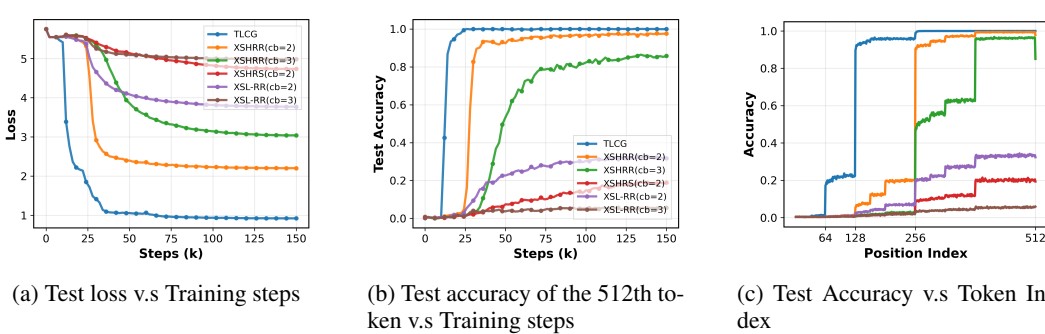

(a) Test loss v.s Training steps

(b) Test accuracy of the 512th to-
ken v.s Training steps

(c) Test Accuracy v.s Token In-
dex

Figure 20: Learning curves and final test performance of a 2-layer model trained on the combined
dataset

PCG variants with more control bits converge more slowly and reach higher loss. The results high-
light that with limited model depth the Transformer prioritizes learning simpler generators first.

# F  CURRICULA

Let the target modulus be $m_{\text{target}}$. The training set contains sequences from $m_{\text{target}}$ as well as from
auxiliary moduli $m_{\text{cur},0}, m_{\text{cur},1}, \ldots$. Curriculum training is defined by a sampling distribution over
these moduli. Each $\alpha_i$ gives the probability of sampling a sequence from $m_{\text{cur},i}$ and the probability
of sampling from $m_{\text{target}}$ is $(1 - \alpha_1 - \alpha_2 - \ldots)$. We vary $\alpha_i$ over training (e.g., exponential decay to
zero) to implement different curriculum schedules. As described in Section 5, we compared fixed-
$\alpha$ curricula with exponentially decaying-$\alpha$ schedules. Here, we extend this analysis by presenting
results with alternative decay functions to evaluate different schedules. We evaluate four curriculum
schedules—cosine decay, exponential decay, linear decay, and step decay—using the XSLRR task
with target modulus $m_2 = 2^{18}$ and auxiliary modulus $m_1 = 2^{16}$. Using a 4-layer, 6-head Transformer
with $d_{\text{model}} = 768$ (batch size=256, context length=512), training begins with sampling weight
$\alpha = 1.0$ from $m_1$ and decays to zero over 100k steps following each schedule. The top row of
Figure 22 shows the sampling probability $\alpha$ over training, and the bottom row reports training and
test accuracy averaged over all token positions in the sequences. Across all schedules, exponential
decay produced the best performance.

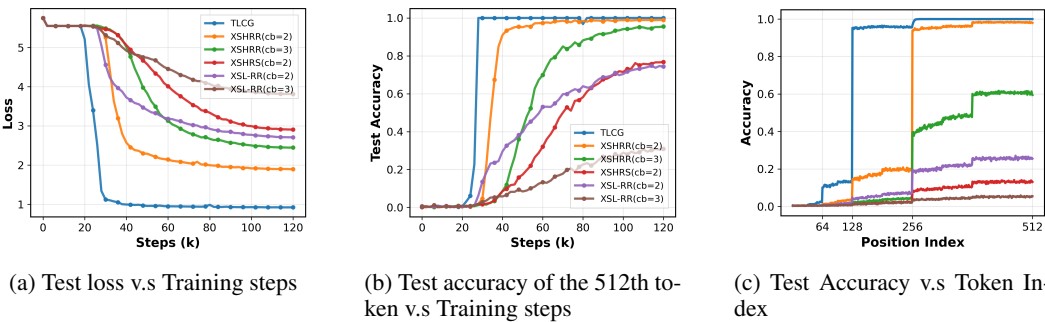

(a) Test loss v.s Training steps

(b) Test accuracy of the 512th token v.s Training steps

(c) Test Accuracy v.s Token Index

Figure 21: Learning curves and final test performance of a 3-layer model trained on the combined dataset

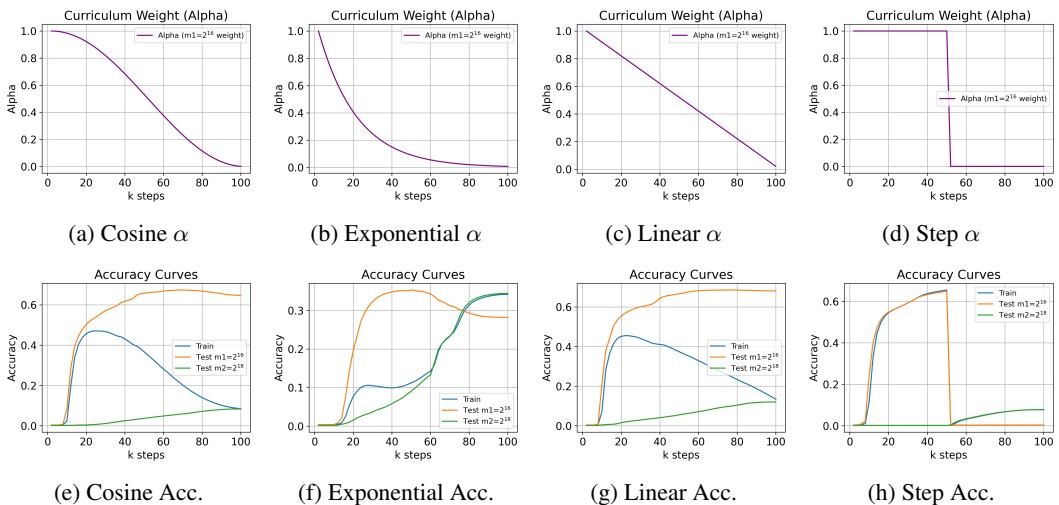

(a) Cosine $\alpha$

(b) Exponential $\alpha$

(c) Linear $\alpha$

(d) Step $\alpha$

(e) Cosine Acc.

(f) Exponential Acc.

(g) Linear Acc.

(h) Step Acc.

Figure 22: Comparison of curriculum schedules. Top row: probability $\alpha$ of sampling from the smaller-modulus dataset over training steps. Bottom row: training and test accuracy on $m_1=2^{16}$ and $m_2=2^{18}$ under each schedule, averaged over all token positions in the sequences.

# G   PRETRAINED INITIALIZATION

To visualize how pretraining affects the evolution of token embeddings, we project the embeddings at Step 0 and at the end of fine-tuning (Step 100,000) onto their first two principal components. The model is initialized from weights trained on XSLRR-16/8 ($m=2^{16}$); since the output vocabulary doubles at XSLRR-18/9 ($m=2^{18}$), the first 256 token embeddings are transferred while the remaining tokens are randomly initialized. Figure 23(a) shows the PCA of the transferred embeddings at pre-trained initialization and end of training, while Figure 23(b) shows the PCA of the randomly initialized embeddings. These plots reveal how pretraining provides a head start: the transferred half begins in an organized configuration and changes slightly during fine-tuning, whereas the randomly initialized half begins unstructured and gradually aligns with the learned embedding space.

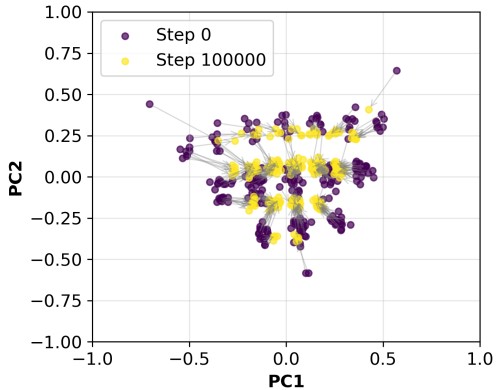 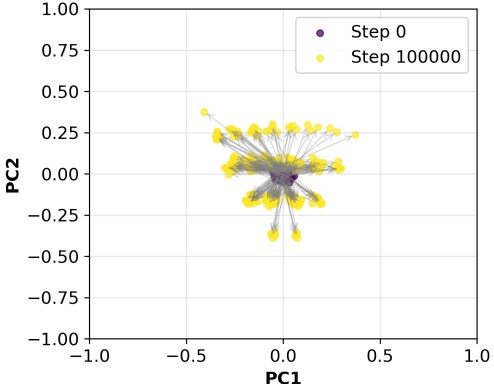

(a) Pretrained initialization: embeddings begin closer to their final configuration.

(b) Random initialization: embeddings start in a unstructured state

Figure 23: PCA of token embeddings at the start (purple) and after fine-tuning (yellow) for a model trained on XSLRR-18/9 $m=2^{18}$. Lines connect the same token across training steps. Pretrained initialization yields embeddings that are already organized.

## H INTERPRETABILITY OF LEARNED REPRESENTATIONS

### H.1 TOKEN EMBEDDINGS

#### H.1.1 XSLRR: TOKEN CLUSTERS

**PC1 and PC2**: Table 1 presents the detailed token clusters for XSLRR-16/8, listing each cluster's canonical binary pattern, its rotationally equivalent variants, and the corresponding tokens assigned to each group.

**PC3 and PC4**: As shown in Figure 24, PC3 aligns with the signed imbalance between even and odd indexed bits ($PC_3(x) \propto (b_0 + b_2 + b_4 + b_6) - (b_1 + b_3 + b_5 + b_7)$), whereas PC4 reflects a weighted contrast across bit positions (approximately $(2b_1 + b_2 + 2b_5 + b_6) - (b_0 + 2b_3 + b_4 + 2b_7)$).

**PC5 and PC6**: In contrast, PC5 and PC6 do not exhibit clean correlations with simple linear functions of the bit positions(Figure 25. Their embeddings show weak and mixed dependencies across bit indices, with no interpretable parity or positional structure analogous to PC3 and PC4. These components therefore warrant further investigation;

The first several principal components each explain a modest but non-negligible portion of the variance. PC1 accounts for 5.1%, PC2 for 4.0%, PC3 for 3.2%, and PC4 for 2.4% of the embedding variance. Together the first four components explain roughly 15% of the total variance, indicating that these directions capture prominent but not exclusive axes of variation in the embedding space. This is consistent with our observation that PC1–PC4 align with simple bit-level statistics, while the remaining variance is distributed across many orthogonal directions needed to preserve token-level distinctions.

**Larger moduli** Figure 26 shows XSLRR token-embedding clusters at larger moduli ($m$). The same rotation-invariant structure observed at $m=2^{16}$ persists as $m$ increases, with clusters aligning to identical zero-run patterns.

#### H.1.2 COMBINED

Figure 27 shows the token embeddings of a model trained on combined datasets, projected onto the first two principal components. Unlike models trained on XSLRR, this model develops a more general, permutation-agnostic grouping of tokens. The embeddings form two broad bands corre-

| Cluster | Pattern | Rotational Equivalent | Tokens |
|---|---|---|---|
| 1 | Z() all bits are `1` | `11111111` | 255 |
| 2 | Z(*) all bits are `0` | `00000000` | 0 |
| 3 | Z(1) only 1 bit is `0` | `01111111` | 127, 191, 223, 239, 247, 251, 253, 254 |
| 4 | Z(2) 2 consecutive `0` | `00111111` | 63, 126, 159, 207, 231, 243, 249, 252 |
| 5 | Z(3) | `00011111` | 31, 62, 124, 143, 199, 227, 241, 248 |
| 6 | Z(4) | `00001111` | 15, 30, 60, 120, 135, 195, 225, 240 |
| 7 | Z(5) | `00000111` | 7, 14, 28, 56, 112, 131, 193, 224 |
| 8 | Z(6) | `00000011` | 3, 6, 12, 24, 48, 96, 129, 192 |
| 9 | Z(7) | `00000001` | 1, 2, 4, 8, 16, 32, 64, 128 |
| 10 | Z(1,1) 2 separated `0` | `01011111` | 95, 125, 175, 190, 215, 235, 245, 250 |
| | | `01101111` | 111, 123, 183, 189, 219, 222, 237, 246 |
| | | `01110111` | 119, 187, 221, 238 |
| 11 | Z(2,1) 2 consecutive `0` and 1 separated `0` | `00101111` | 47, 94, 121, 151, 188, 203, 229, 242 |
| | | `00110111` | 55, 110, 115, 155, 185, 205, 220, 230 |
| | | `00111011` | 59, 103, 118, 157, 179, 206, 217, 236 |
| | | `00111101` | 61, 79, 122, 158, 167, 211, 233, 244 |
| 12 | Z(3,1) or Z(2,2) | `00010111` | 23, 46, 92, 113, 139, 184, 197, 226 |
| | | `00011011` | 27, 54, 99, 108, 141, 177, 198, 216 |
| | | `00011101` | 29, 58, 71, 116, 142, 163, 209, 232 |
| | | `00100111` | 39, 57, 78, 114, 147, 156, 201, 228 |
| | | `00110011` | 51, 102, 153, 204 |
| 13 | Z(4,1) or Z(3,2) | `00001011` | 11, 22, 44, 88, 97, 133, 176, 194 |
| | | `00001101` | 13, 26, 52, 67, 104, 134, 161, 208 |
| | | `00010011` | 19, 38, 49, 76, 98, 137, 152, 196 |
| | | `00011001` | 25, 35, 50, 70, 100, 140, 145, 200 |
| 14 | Z(5,1) or Z(4,2) or Z(3,3) | `00000101` | 5, 10, 20, 40, 65, 80, 130, 160 |
| | | `00001001` | 9, 18, 33, 36, 66, 72, 132, 144 |
| | | `00010001` | 17, 34, 68, 136 |
| 15 | Z(1,1,1) | `01010111` | 87, 93, 117, 171, 174, 186, 213, 234 |
| | | `01011011` | 91, 107, 109, 173, 181, 182, 214, 218 |
| 16 | Z(2,1,1) | `00101011` | 43, 86, 89, 101, 149, 172, 178, 202 |
| | | `00101101` | 45, 75, 90, 105, 150, 165, 180, 210 |
| | | `00110101` | 53, 77, 83, 106, 154, 166, 169, 212 |
| 17 | Z(3,1,1) or Z(2,2,1) | `00010101` | 21, 42, 69, 81, 84, 138, 162, 168 |
| | | `00100101` | 37, 41, 73, 74, 82, 146, 148, 164 |
| 18 | Z(1,1,1,1) | `01010101` | 85, 170 |

Table 1: XSLRR-16/8 Model Token Clusters with Rotation-Based Structures

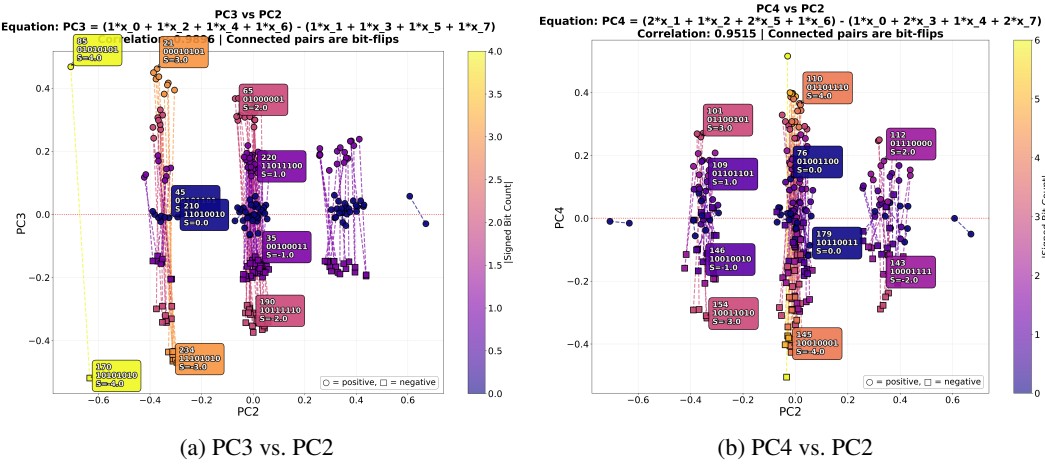

(a) PC3 vs. PC2       (b) PC4 vs. PC2

Figure 24: PC3 and PC4 principal component structure. Connected points denote bit-flip pairs.

sponding to even and odd tokens. Along PC2, tokens are roughly ordered from top to bottom by increasing numbers of 1-bits.

## H.2 ATTENTION PATTERN

To analyze how attention spans evolve across the network, Figure 28 shows the distribution of token distances for the top-8 most-attended keys per query, averaged over all positions and heads in each layer (Distances of $q - k = 0$ are excluded because self-attention peaks there and would dominate

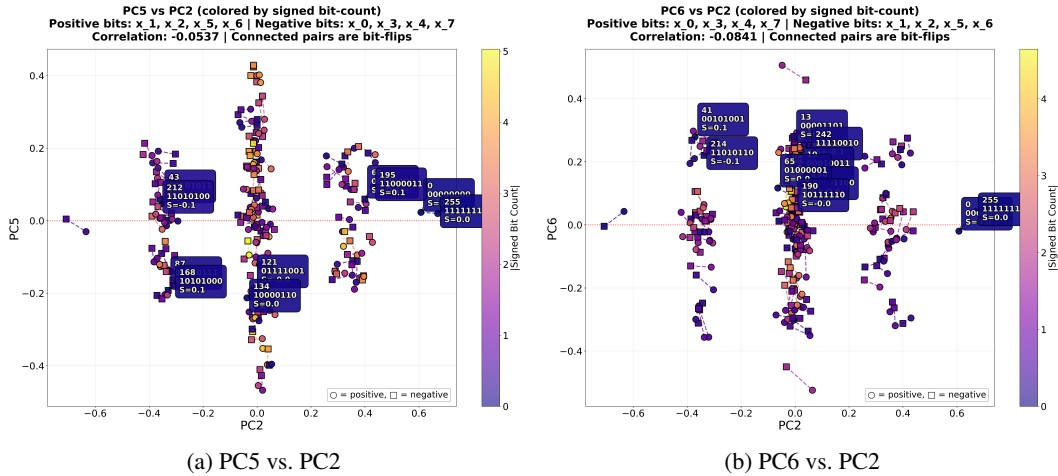

(a) PC5 vs. PC2

(b) PC6 vs. PC2

Figure 25: PC5 and PC6 principal components. Connected points denote bit-flip pairs.

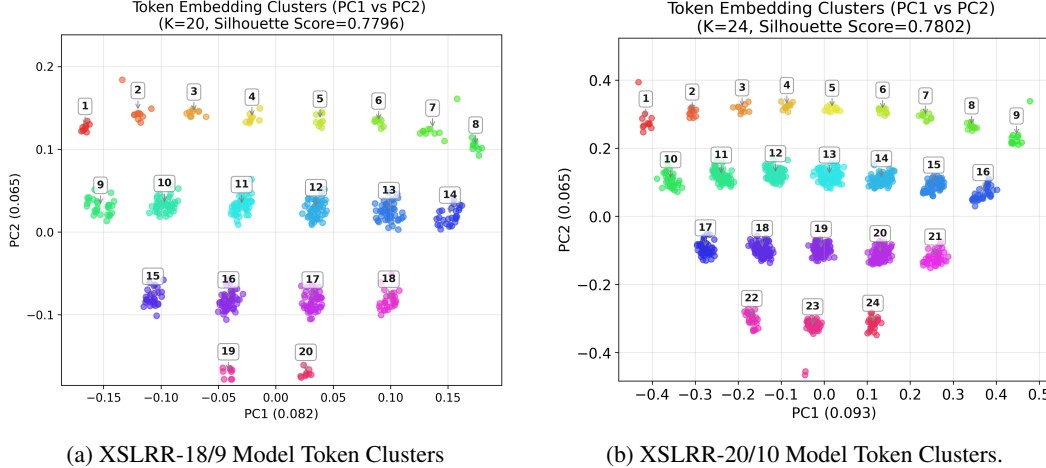

(a) XSLRR-18/9 Model Token Clusters

(b) XSLRR-20/10 Model Token Clusters.

Figure 26: PCA analysis of token embeddings. The same rotation-invariant grouping structure persists across moduli, with clusters consistently aligned to zero-run patterns.

the scale). In the first layer, attention is dominated by long-range periodic connections, with strong peaks at powers of two (64, 128, 256), revealing that the model has discovered the underlying bit-periodicity of the generators. By the later layers, attention shifts toward shorter token distances, indicating that prediction increasingly relies on local context once the global recurrence has been inferred.

## H.3 HEAD SPECIALIZATION ABLATION

To test whether individual attention heads specialize on particular aspects of the prediction task, we performed a single-head ablation study. We zeroed out the V slice of each specified head, effectively removing that head's contribution while leaving the rest of the network unchanged. We evaluated every head in every layer and measured the resulting accuracy drop at the 512-th token relative to the unablated baseline. As shown in Figure 29, head specialization emerges clearly in the last three layers: for example, at the last layer, Heads 0 and 6 minimally affects XSLRR but substantially reduces accuracy on truncated LCG, whereas Heads 3 and 5 strongly affect XSLRR but not truncated LCG. These results indicate that late-layer attention heads develop task-specific roles, with some focusing on full-state PCGs and others adapting to truncated outputs.

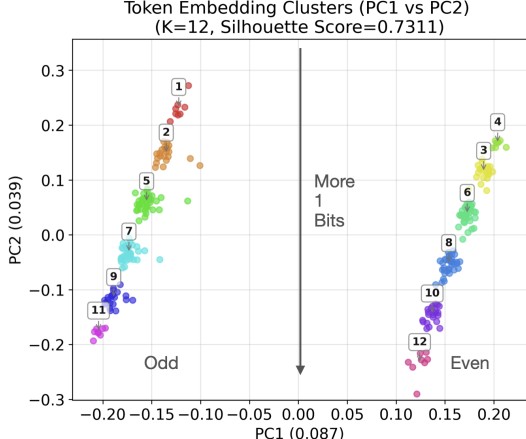

Figure 27: When trained on combined datasets, the model develops a more general grouping of tokens. The visualization shows token embeddings projected onto the first two principal components.

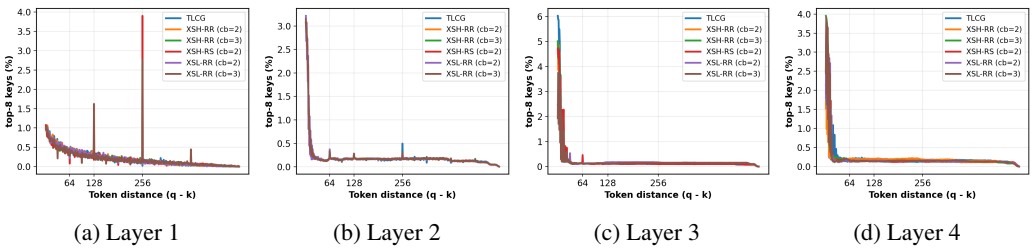

| (a) Layer 1 | (b) Layer 2 | (c) Layer 3 | (d) Layer 4 |

Figure 28: Token-distance distribution of the top-8 attended keys at each Transformer layer for a model trained on the combined dataset.

# I  USE OF LARGE LANGUAGE MODELS

Portions of the text were refined with the assistance of Large Language Models to improve grammar.

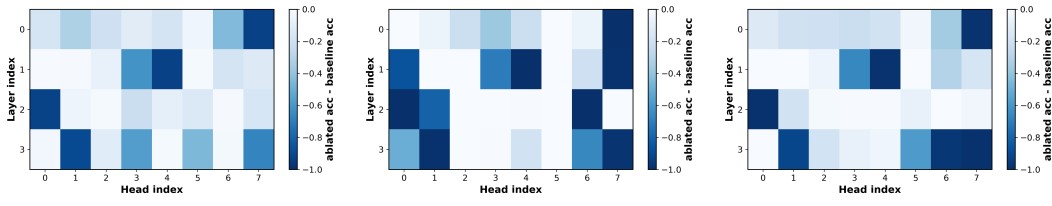

(a) XSLRR PCG at position 512.  (b) Truncated LCG at position 512.  (c) XSHRR PCG at position 512.

Figure 29: **Single-head ablation.** Accuracy drop at the 512th token when zeroing the V-slice of each attention head. Darker colors indicate a larger decrease in accuracy relative to the baseline. The last two or three layers exhibit stronger head specialization, with some heads (e.g., Head 6) affecting truncated LCG but not XSLRR, and others (e.g., Heads 3 and 5) affecting XSLRR but not truncated LCG.

