# OpenReview forum: "Learning Pseudorandom Numbers with Transformers: Permuted Congruential Generators, Curricula, and Interpretability"
_ICLR.cc/2026/Conference — ICLR 2026 Poster_

### Official Review · Reviewer_2asJ · 2025-10-30

**Soundness:** 4
**Presentation:** 4
**Contribution:** 3
**Rating:** 8
**Confidence:** 3

**Summary:**

*Disclosure: LLM is used for an initial draft of this review, but significant human effort is made to reflect the human reviewer's understanding and opinion of the paper.*

This paper investigates the ability of Transformer models to learn and predict sequences from Permuted Congruential Generators (PCGs), a family of pseudo-random number generators (PRNGs) more complex than standard Linear Congruential Generators (LCGs) due to their use of bitwise permutations (shifts, XORs, rotations).

The authors demonstrate several key findings:
1.  **In-Context Learning:** Transformers can successfully perform in-context prediction of sequences from various PCG variants, generalizing to unseen generator parameters (a, c). This capability holds even when the output is severely truncated (e.g., to a single bit). A single model trained on a combined dataset of different PCG variants can also learn to identify and predict sequences from all of them.
2.  **Training Strategy:** Training models on large moduli directly is infeasible due to prolonged stagnation. The authors show that mixing in data from smaller moduli or using a pretrained model are important for overcoming this stagnation and successfully training large-modulus models.
3.  **Scaling Law:** The authors identify an empirical scaling law: the number of in-context sequence elements L required for near-perfect prediction grows with the modulus m as $L \propto \sqrt{m}$. This is a steeper requirement than for LCGs ($L \propto m^{0.25}$).
4.  **Interpretability:** By analyzing the token embedding layer, the paper reveals that the model spontaneously organizes representations based on rotation-invariant features (specifically, the number and arrangement of contiguous zero-runs in the tokens' binary form). This learned structure explains why representations transfer effectively from smaller to larger moduli.

**Strengths:**

- PCGs are practically relevant and designed to be statistically hard to predict. This work effectively establishes a new, ML-based approach to cryptanalysis and could evolve into a practical benchmark or tool for evaluating the security of other PRNGs. The experiments are throughout and comprehensive.

- The paper provides new insights into the expressivity of transformers, demonstrating they can model surprisingly complex, non-linear bitwise operations (not just simple arithmetic). The ablations offer a concrete analysis of the importance of pre-training and curriculum learning.

- The discovery of the $L \propto \sqrt{m}$ scaling law is intriguing, potentially related to known attacks such as Sweet32 attacks.

**Weaknesses:**

- The data efficiency of the proposed method is not great. It is unclear that if the accuracy mainly comes from memorizing similar patterns.

- (Minor) The findings around interpretability is not particularly novel, as previous studies on grokking and probing revealed similar geometric structures. The paper will be even better with a more in-depth mechanistic interpretability analysis.

**Questions:**

- I would like to see a power law fit on the dataset size. Is the dataset size (# of sequences to be seen) needed linear w.r.p. to m?

- Is the base-64/128/256 training empirically better than bit-wise tokenization? Would be good if we have some justifications.

---

> ### Author Response · Authors · 2025-11-24
> **Dataset Size Scaling; Justification for Base-256 Tokenization; Data Efficiency and Memorization Concerns**
>
> We thank the reviewer for recognizing our main contributions, as well as for the helpful suggestions on improving clarity and analysis.
>
> **Dataset Size Scaling**
>
> Producing a power-law plot would require identifying the minimal dataset sizes $n_a$ and $n_c$ for each modulus $m$, which in practice means running a learning rate and weight decay sweep for every dataset size. As $m$ increases, the required training steps also increase, and optimization becomes more sensitive to hyperparameters and stochasticity. This makes such an analysis greatly exceeds the computational budget for this project. Additionally, in our current experiments, simply increasing dataset size is not sufficient for large moduli (as shown in Figure 17 in Appendix D); curriculum learning is necessary for the model to train. So a fair comparison requires careful design and analysis of the experiments, in addition to substantially more of them. We view this scaling as a valuable direction when additional compute becomes available.
>
> **Base-64/128/256 Tokenization vs. Bit-Wise Tokenization**
>
> Base-64/128/256 tokenization is a practical trade-off between context length and vocabulary size. With bit-wise tokenization, an 8-bit output becomes 8 tokens, which increases the context length by a factor of 8. Since attention cost scales as $O(L^2)$, this results in roughly $64\times$ more attention memory and compute, making training substantially more expensive. Conversely, for generators with wider outputs (e.g., a truncated LCG with 15-bit outputs), using a full vocabulary of size $2^{15} = 16384$ is inefficient and harder to train, while grouping bits into base-256 tokens keeps the vocabulary small at the cost of doubling the context length. In practice, the base-256 schemes provide a balanced compromise that keeps both sequence length and vocabulary size within a tractable range. We have added this justification to section 2.3 in the main text and to the truncated LCG section in appendix B.
>
> **Data Efficiency and Memorization Concerns**
>
> The model is not simply memorizing patterns. A key piece of evidence is that it generalizes to unseen recurrence parameters $(a,c)$, which cannot be achieved through memorization alone. The data inefficiency (in terms of required in-context inputs) is more likely due to the model discovering a suboptimal algorithmic strategy. In our experiments, larger models learn substantially more data-efficient solutions.
>
> **Full Mechanistic Interpretability**
>
> A full mechanistic interpretation of the model’s algorithm is an important direction for future research. Given its technical complexity, we view this as best suited for a dedicated follow-up project.

---

### Official Review · Reviewer_i5nj · 2025-11-01

**Soundness:** 3
**Presentation:** 3
**Contribution:** 2
**Rating:** 4
**Confidence:** 3

**Summary:**

The paper investigates whether Transformer models can perform in-context next-state prediction on sequences generated by pseudorandom congruential generators (PCGs).
The authors consider four PCG variants (TLCG, XSLRR, XSHRR, XSHRS) and train models both on per-generator datasets and on a combined dataset.
The results show that Transformers not only learn each generator’s rule but also can infer which generator produced a given sequence.
The authors further study how performance scales with the modulus $m$, dataset size, and model size.
The experimental results find that curriculum learning with data-mixing strategies is effective for large moduli.
Finally, a PCA of the embedding matrix suggests the model leverages structural statistics such as the number of zeros and zero-run patterns to support in-context reasoning.

**Strengths:**

**S1.** The paper is clearly written, and the figures are well-designed.

**S2.**  The authors conduct extensive experiments to support the paper’s claims, and the results convincingly substantiate those claims.

**Weaknesses:**

**W1.**
The PCG setup, where a hidden state $s_i$ evolves and the observation $x_i$ is produced via a deterministic function $f$, is conceptually close to HMMs and finite-state automata.
To my knowledge, there is already substantial work probing Transformers’ capability on learning HMM/automata-like processes [1,2].
I think the paper should clarify what is genuinely new here versus what might already follow from known results on those finite-state structures.

**W2.**
The PCA-based analysis suggesting the model learns and exploits a “zero-run” pattern is intriguing, but it does not provide a full understanding of how Transformers actually perform in-context learning on PCG instances.
Given that PCG sequences are challenging to analyze, a complete theory may be out of scope; however, I feel the analysis remains largely phenomenological.
As a result, the contribution read as “pick some convenient task, train a Transformer under various settings, and show that it exhibits in-context learning,” rather than explaining the underlying mechanism.

---
[1] Hu, Jiachen, Qinghua Liu, and Chi Jin. "On limitation of transformer for learning hmms." arXiv preprint arXiv:2406.04089 (2024).

[2] Liu, Bingbin, et al. "Transformers learn shortcuts to automata." arXiv preprint arXiv:2210.10749 (2022).

**Questions:**

**Q1.**
In lines 263-264, the authors claim that a 1-layer model suffices to solve small-modulus PCG instances.
However, to my knowledge, prior ICL work on Markov-chain-style tasks suggests that at least two layers are required to implement induction heads and achieve ICL [3].
Although PCG is not a Markov chain, it is more complex.
Could you clarify how PCG can be solved with a shallower model than Markov-chain-style tasks?

**Q2.**
I personally find the PCA analysis interesting: the first axis aligns with the total number of zero bits, and the second with the number of zero runs.
Have you examined additional principal components (e.g., 3-component)?
If so, does the third component also align with interpretable statistics?

---

[3] Ekbote, Chanakya, et al. "What One Cannot, Two Can: Two-Layer Transformers Provably Represent Induction Heads on Any-Order Markov Chains." *arXiv preprint arXiv:2508.07208* (2025).

---

> ### Author Response · Authors · 2025-11-24
> **Comparison with Hidden Markov Models and Automata; Analysis of Higher-Order Principal Components of the Embedding Layer**
>
> **Comparison with Hidden Markov Models and Automata**
> - Hidden Markov Models (HMMs): HMMs are governed by probabilistic transition and emission processes. In the works cited[1,2,3,4], these are determined by probability matrices that lack any underlying algorithmic structure. The models were trained on many examples of these transitions and emissions in order to infer and memorize the corresponding matrices. In contrast, in PCGs, both the transition and emission are defined by a rich deterministic algorithm. Our models are trained to discover the underlying algorithm rather than to memorize transition or emission tables.
>
> - Automata: In the cited automata work[5], transitions are driven by external input symbols. The task takes the form: given an initial state $s_0$ and an input sequence $x_{1:T}$, can a Transformer reconstruct the hidden states $s_{1:T}$, where each transition $s_t \rightarrow s_{t+1}$ is determined by both the current state $s_t$ and the current input symbol $x_t$?
> In this setting, the model is effectively trained on many transition examples in order to learn the transition rule from the current state and input symbol to the next state $(x_t, s_t) \mapsto s_{t+1}$. The difficulty in the automata task lies in inferring the transition rule from incomplete observations of these transitions.
> In contrast, PCGs have no external input that control the transition; the recurrence is fixed, and state-only. The outputs $x_t$ are derived from the internal state $s_t$. The corresponding task is therefore: Given PRNG outputs $x_{0:T}$, can a Transformer predict $x_{T+1}$? The difficulty lies in generalizing to completely unseen transition parameters $a$ and $c$.
>
> As summarized above, the PCG setting is fundamentally different from prior HMM and automata works, leading to two key experimental distinctions:
>
> - Generalization. In the HMM and automata tasks studied in prior work, models cannot generalize to unseen transitions. As shown in Figure 4 (right) of [2], when test prompts are drawn from previously unseen transition matrices, in-context learning fails to extrapolate. To our knowledge, generalization to unseen transitions is not considered in the automata works. The main point of our work is that the model successfully generalizes in-context to the case with unseen parameters $(a, c)$.
> - State and output space sizes. Existing HMM-based works operate with very small state and output spaces (typically $\leq 64$). Similarly, in automata settings, the largest state space considered is $120$. In contrast, the PCGs we study involve state and output spaces of size $2^{22}$ and $2^{11}$, respectively.
>
> Therefore, prior works address a substantially different problem involving small finite-state systems, whereas our goal is to study ICL generalization to unseen transition parameters. We have added this clarification to the Related Work section of the revision.
>
> **Analysis of Higher-Order Principal Components of the Embedding Layer**
>
> We thank the reviewer for this question and motivating us to look at higher principal components.
> We extended our analysis to the first six principal components and found that PC3 and PC4 are also interpretable.
>
> PC3 correlates strongly with the signed difference between sums of bits in even and odd positions:
>
> $\mathrm{PC}_3(x) \propto (b_0 + b_2 + b_4 + b_6) - (b_1 + b_3 + b_5 + b_7),$
> where $b_i$ denotes the i-th bit of the token.
>
> PC4 corresponds to a weighted signed difference of bit positions: $\mathrm{PC}_4 \propto (2b_1 + b_2 + 2b_5 + b_6) - (b_0 + 2b_3 + b_4 + 2b_7)$.
>
> PC5 and PC6 do not exhibit clean correlations with simple linear functions of the bits, and thus would require further analysis.
> We include these additional PCA visualizations and correlations in the appendix H.1.1.
>
> **References**
>
> [1] Wei et al., Why Do Pretrained Language Models Help in Downstream Tasks? An Analysis of Head and Prompt Tuning. NeurIPS 2021.
>
> [2] Xie et al., An Explanation of In-context Learning as Implicit Bayesian Inference. ICLR 2022.
>
> [3] Dai et al., Pre-trained Large Language Models Learn Hidden Markov Models In-context. NeurIPS 2025.
>
> [4] Hu et al., "On limitation of transformer for learning hmms." arXiv preprint arXiv:2406.04089 (2024).
>
> [5] Liu et al., "Transformers learn shortcuts to automata." arXiv preprint arXiv:2210.10749 (2022).

---

> ### Author Response · Authors · 2025-11-24
> **Why PCG Prediction Does Not Require Two Layers; Motivation and Justification for Our Contribution**
>
> **Why PCG Prediction Does Not Require Two Layers**
>
> The cited work of Ekbote et al. [1] proves that a two-layer Transformer can represent any $k$-th order Markov process. However, this result does not establish that such architectures are necessary for all Markov-chain tasks. In principle, certain Markov processes may admit solutions that do not rely on an induction head. Sanford et al. [2] show that a one-layer, single-head Transformer can represent 1-gram processes if its hidden dimension grows exponentially relative to a two-layer model. This implies that a one-layer model can solve the 1-gram task, but only in a highly inefficient parameter regime.
>
> PCG is a first-order deterministic Markov chain. In contrast to the probabilistic $k$-th-order Markov task considered in [1], where the model must locate a previous occurrence of the current $k$ tokens and copy their subsequent token as output, PCG prediction does not rely on an induction head or any match-and-copy mechanism. For PCGs, the attention layer only needs to aggregate outputs from a fixed set of positions without any content matching or lookup.
>
> **Motivation and Justification for Our Contribution**
>
> Our goal is to probe the algorithmic capabilities and limitations of Transformers by testing it on data with pseudorandom structure. We believe this is a general and deep question that should receive focused attention. A wide open area for this is the controlled setting of PRNG prediction illustrating what hidden rules the model can extract and what factors shape its generalization. From the standpoint of AI for cryptography, our work also tests how far transformers remain from learning cryptographically secure PRNGs or encryption.
>
> As listed as a weakness, there is concern that our contribution appears as “pick some convenient task, train a Transformer, and show that it exhibits ICL.” We regret that our presentation did not clearly convey our motivation and we revise the explanation of motivations in our paper to properly present this perspective.
>
> Through this work, we find that Transformers outperform classical cracking algorithms in several respects: they can learn multiple PRNG variants and generalize without knowing the multiplier $a$, but they remain far less input efficient and require substantially more data and context to succeed. We recognize that a complete reverse-engineering of the model’s internal algorithm is an important direction. Nevertheless, we feel that the discoveries reported in our paper -- the fact that PCGs can be successfully jointly learned and generalized, the discoveries of the $\sqrt{m}$ scaling law, crucial importance of curriculum, and novel rotationally invariant structures in embedding matrices -- are substantial and surprising enough that they warrant publication in ICLR. A deeper, more comprehensive mechanistic interpretability study is thus beyond the scope of the current work and ideal for a future project.
>
> **References**
>
> [1] Ekbote et al. What One Cannot, Two Can: Two-Layer Transformers Provably Represent Induction Heads on Any-Order Markov Chains, 2025.
>
> [2] Sanford et al., One-layer transformers fail to solve the induction heads task, 2024.

---

### Official Review · Reviewer_yPnh · 2025-11-03

**Soundness:** 2
**Presentation:** 3
**Contribution:** 1
**Rating:** 4
**Confidence:** 3

**Summary:**

This paper scrutinizes a problem in learning PCG-generated integral sequences in-context and auto-regressively with Transformers, where PCG stands for Permuted Congruential Generators, a class of PRNGs. Empirically, a scaling law for the number of initial elements required to achieve at least 90% accuracy for unseen parameters ($a$ and $c$) is observed/proposed as ~$0.5\times m^{0.5}$. Moreover, this work offers a wide range of analyses about training methods (e.g., curriculum learning, data mixing) and model representations (e.g., PC analyses of token embeddings and similarity analyses of intermediate features).

**Strengths:**

1. The paper is well written. The problem setting and experimental results are clearly presented.
2. I find most of the findings of the paper very interesting (especially the benefits of curriculum learning with the help of a smaller modulus, and the principal component analyses of embedding vectors). Even though the problem scale (e.g., number of bits required to represent the problem) studied here is quite smaller than the practically used PCGs, I believe it will be a stimulating example for many researchers who are interested in the (in)capability of a Transformer model and its learning dynamics. Indeed, it projects several immediate future works. Can we mathematically prove that/why/how a Transformer learn PCG-generated sequences (or, generalized modular arithmetics in general) in-context with a gradient-based optimizer? How far can we get with constant-size models? How about logarithmically-scaled models?

**Weaknesses:**

1. The observation that transformers can learn PCG-generated sequences auto-regressively may not be a very surprising finding. In fact, the studied problem is not that random since its scale is too small to pass a collection of empirical randomness tests (e.g., BigCrush). Hence, the problem possesses its own auto-regressive nature by its definition, which can be effectively solved with Transformers up to some extent.
2. Indeed, there has already been a huge literature on in-context learning with Transformers in solving problems with a structure of a hidden Markov model (HMM) [1--6]. However, although the PCG seems to be a special instance of HMM, I cannot find a discussion about any of these previous works.
3. While being interesting, the principal component analysis seems erroneous and incomplete; I am particularly suspicious about the ‘clustering’ of the embedding vectors. Do they really cluster together? Aren’t they just getting close because of the projection onto a two-dimensional subspace? Since the token embedding matrix is also the weight matrix of the final linear readout (due to weight sharing of GPT-based models), I don’t think the model is learning a “rotationally invariant” features; they must be distinguishable, but the separation between “rotationally invariant” tokens (in binary representation) is just happening in some other subspace(s) orthogonal to the PC1 and PC2. Being a fully empirical work, I believe the paper must provide a complete analysis of the embedding vectors, discovering on which subspace a particular feature of token embedding vectors is getting separated (e.g., PC1 separates tokens based on the number of zero-bits, PC2 separates tokens based on the number of zero-runs, PC3 separates …).

---

References:

[1] Wei et al., Why Do Pretrained Language Models Help in Downstream Tasks? An Analysis of Head and Prompt Tuning. NeurIPS 2021.

[2] Xie et al., An Explanation of In-context Learning as Implicit Bayesian Inference. ICLR 2022.

[3] Hu et al., On Limitation of Transformer for Learning HMMs. arXiv preprint 2024.

[4] Zhou et al., Transformers learn variable-order Markov chains in-context. arXiv preprint 2024.

[5] Dai et al., Pre-trained Large Language Models Learn Hidden Markov Models In-context. NeurIPS 2025.

[6] Hao et al., Transformers as Multi-task Learners: Decoupling Features in Hidden Markov Models. arXiv preprint 2025.

**Questions:**

1. Why does the observed separation between the learned token embedding vectors really help the model to achieve a near-perfect in-context learning? Yes, I agree that the analysis itself is quite intriguing and visually satisfying, and it gives us some hint about why pre-training enhances the training efficiency for large moduli $m$. However, I don’t think it is a complete analysis of the mechanism of model predictions. One possible way to complete the analysis is to design a pseudo-code that is possibly learned by the model to make correct auto-regressive predictions of PCG-generated sequences.
1. I hope a future revision of the paper will contain some more in-depth implications of predicting pseudo-random numbers with a deep neural network.

---

> ### Author Response · Authors · 2025-11-23
> **On the Comparison with Hidden Markov Models; Analysis of Higher-Order Principal Components of the Embedding Layer**
>
> **Comparison with Hidden Markov Models**
>
> HMMs are governed by probabilistic transition and emission processes. In the works cited, these are determined by probability matrices that lack any underlying algorithmic structure. The models were trained on many examples of these transitions and emissions in order to infer and memorize the corresponding matrices. In contrast, in PCGs, both the transition and emission are defined by a rich deterministic algorithm. Our models are trained to discover the underlying algorithm rather than to memorize transition or emission tables. This difference leads to two key experimental distinctions:
>
> - Generalization. In the HMM tasks studied in prior work, models cannot generalize to unseen transition or emission matrices. As shown in Figure 4 (right) of [2], when test prompts are drawn from previously unseen transition matrices, in-context learning fails to extrapolate. The main point of our work is that the model successfully generalizes in-context to the case with unseen parameters $(a, c)$.
> - State and output space sizes. Existing HMM-based works operate with very small state and output spaces: in [1,2], both hidden state size $|\mathcal{S}|$ and output size $|\mathcal{X}|$ are only 10, and in [3] the largest setting is 64. In contrast, the PCGs we study involve state and output spaces of size $|\mathcal{S}| = 2^{22}$ and $|\mathcal{X}| = 2^{11}$, respectively.
>
> Therefore, while formally PCGs can be thought of as a deterministic version of HMMs, the scope of prior work in this subject is substantially different from our study. We have added this discussion to the Related Work section of the revision.
>
> **Analysis of Higher-Order Principal Components of the Embedding Layer**
>
> We agree with the reviewer that higher order principal components can of course distinguish the embedding vectors. The point of our results was to show that PC1 and PC2 do demonstrate this rotationally invariant clustering, which is a new kind of emergent structure in the embedding vectors that has not been observed in prior work. In the revision, we clarify that terms like “clusters” and “rotation invariance” refer only to structure observed in the projected principal components, not to literal collapsing of embeddings in the full space.
>
> As suggested by the reviewer, we have further extended our PCA analysis to quantify the variance explained by each component and to examine higher-order directions (PC3–PC6). We find PC3 and PC4 correspond to finer bit-level features.
> PC3 captures a signed even–odd bit imbalance:
>
> $\mathrm{PC}_3(x) \propto  (b_0 + b_2 + b_4 + b_6)-(b_1 + b_3 + b_5 + b_7)$, where $b_i$ denotes the i-th bit of the token.
>
> PC4 corresponds to a weighted bit-difference pattern. Approximately:
>
> $\mathrm{PC}_4 \propto (2b_1+b_2+2b_5+b_6)-(b_0+2b_3+b_4+2b_7)$.
>
>
> PC5 and PC6 do not exhibit clean correlations with simple linear functions of the bits, and thus would require further analysis. Detailed results are provided in Appendix~H.1.1.
>
> We thank the reviewer for these suggestions, which have strengthened our results.
>
>
> **References**
>
> [1] Wei et al., Why Do Pretrained Language Models Help in Downstream Tasks? An Analysis of Head and Prompt Tuning. NeurIPS 2021.
>
> [2] Xie et al., An Explanation of In-context Learning as Implicit Bayesian Inference. ICLR 2022.
>
> [3] Dai et al., Pre-trained Large Language Models Learn Hidden Markov Models In-context. NeurIPS 2025.

---

> ### Author Response · Authors · 2025-11-23
> **On How Token Embedding Geometry Facilitates Algorithmic Generalization; Implications of Predicting Pseudo-Random Numbers**
>
> **Why does the observed separation between the learned token embedding vectors really help the model to achieve a near-perfect in-context learning?**
>
> Our embedding analysis is intended as an initial step toward understanding how the model organizes PRNG states. The close correspondence between the learned embedding geometry and the generator’s underlying structure suggests that the model leverages the patterns encoded in the top principal components. Importantly, as we increase the modulus, these structures persist and extend, suggesting the learned organization is algorithmic.
>
> A complete mechanistic interpretation of the model’s algorithm is an important direction of future research, but it is technically substantial and beyond the scope of this paper, which in our view already contains a significant number of non-trivial results. It is worth mentioning that, to the best of our knowledge, there is no classical attack that can recover PCG sequences when the multiplier $a$ is unknown. This makes it difficult to propose a hypothesis for the exact computation the model performs.
>
> **In-depth implications of predicting pseudo-random numbers with a deep neural network**
>
> In the revised version, we have added a dedicated Discussion section that clarifies both the implications and the limitations of our empirical study.
>
> From a broad perspective, we believe the general question of learning pseudorandom structure in data is important as a way to test the limits of capabilities of various deep learning paradigms.
>
> Predicting PRNG outputs thus provides a controlled way to study the algorithmic capabilities and complexity limits of Transformers. It shows how intricate the hidden rules are that the model can extract and what factors constrain its ability to approach practical PRNGs. As the difficulty of the generators increases, we find that both curriculum learning and sufficient context length become essential for successful generalization. Compared with classical attacks, Transformers can learn multiple PRNG variants and extrapolate to unseen parameters, but their performance declines as the modulus and state complexity grow. Studying simpler PRNGs is also an important step before asking whether deep models can learn anything about cryptographically secure generators.

---

### Author Response · Authors · 2025-11-24
**Summary of Revision**

We summarize the main concerns raised by the reviewers and how we address them in the revision.

**Novelty vs. Hidden Markov Model (HMM) work:** Two reviewers initially gave low contribution ratings due to perceived similarity with HMM work and asked us to clarify how our setting differs. We clarify that those studies learn or memorize transition tables in small finite-state systems(typically $\leq 64$), while Transformers in our setting must discover a deterministic algorithm and generalize to unseen generator parameters $(a,c)$ over state spaces of size $2^{22}$.

**Principal component analysis was incomplete:** Reviewers suggested analyzing higher-order principal components. We extended the analysis up to PC6, quantified explained variance, and showed that PC3/PC4 also correspond to interpretable bit-level patterns.

**Complete algorithm and contributions:**
The reviewer asked for a complete algorithm implemented by the model, and questioned whether this limits our contribution. A full algorithmic solution is currently out of reach (and classically unavailable for unknown multiplier $a$ for PCGs). Our contribution is to identify algorithmic behaviors: (1) Transformers can learn multiple PRNG variants simultaneously, (2) they generalize to unseen generator parameters $(a,c)$, (3) the required context length scales with the modulus $m$ as $\sqrt{m}$, (4) curriculum is required for larger $m$, and (5) the learned embedding space exhibits a novel structure that persists and expands with $m$.

**Other questions:**
Reviewers also asked about implications, minimal depth, dataset size scaling, tokenization, and memorization. We addressed these individually in the response.

---

### Meta-Review · Area_Chair_k8MW · 2026-01-07

**Summary:**

Strengths

- Well written paper (yPnH, i5nj)
- Extensive experiments (i5nj)
- PCGs are a great way to study transformer learning dynamics (i5nj, yPnh)

Weaknesses

- No discussion of the connections between PCGs and HMMs (yPnh, i5nj)
- Limited analysis of the PCA results (yPnh, i5nj, 2asJ)

**Reviewer Concerns:**

The authors addressed both concerns:

- Good discussion on the similaries between this work and previous HMM learning works.
- Additional analysis on PCA dimensions.

There are no outstanding issues as far as I can tell.

**Reviewer Scores:**

I think all reviewers would have raised their scores.

---

### Decision · Program_Chairs · 2026-01-26

Accept (Poster)